# Canonical Rank Adaptation: An Efficient Fine-Tuning Strategy for Vision Transformers

Lokesh Veeramacheneni [1]   Moritz Wolter [1]   Hilde Kuehne [2 3]   Juergen Gall [1 4]

## Abstract

Modern methods for fine-tuning Vision Transformers, such as Low-Rank Adaptation (LoRA) and its variants, demonstrate impressive performance. However, these methods ignore the high-dimensional nature of Multi-Head Attention (MHA) weight tensors. To address this limitation, we propose Canonical Rank Adaptation (CaRA). CaRA leverages tensor mathematics, first by tensorising the transformer into two different tensors: one for projection layers in MHA and the other for feed-forward layers. Second, the tensorised formulation is fine-tuned using the low-rank adaptation in the Canonical-Polyadic Decomposition (CPD) form. Employing CaRA efficiently minimises the number of trainable parameters. Experimentally, CaRA outperforms existing Parameter-Efficient Fine-Tuning (PEFT) methods in visual classification benchmarks such as the Visual Task Adaptation Benchmark (VTAB)-1k and the Fine-Grained Visual Categorization (FGVC) benchmark.

## 1. Introduction

While Vision Transformer (ViT) architectures demonstrate remarkable performance on a wide set of tasks, from classification (Dosovitskiy et al., 2021) to semantic segmentation (Xie et al., 2021), they are data-hungry (Raghu et al., 2021) and pre-trained ViTs are required to employ them in general downstream tasks with limited data. This is typically done by full fine-tuning (FT), which is inefficient as networks increase in scale. To circumvent this issue, Parameter-Efficient Fine-Tuning (PEFT) methods, first introduced by Houlsby et al. (2019), fine-tune only a subset of additional

[1]University of Bonn [2]Tuebingen AI Center [3]MIT-IBM Watson AI Lab [4]Lamarr Institute for Machine Learning and Artificial Intelligence. Correspondence to: Lokesh Veeramacheneni <lokiv@uni-bonn.de>.

*Proceedings of the $42^{nd}$ International Conference on Machine Learning*, Vancouver, Canada. PMLR 267, 2025. Copyright 2025 by the author(s).

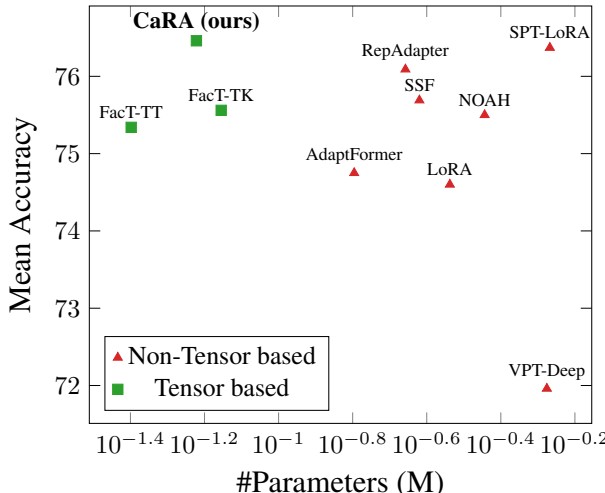

*Figure 1.* Mean accuracy on the VTAB-1k benchmark versus the number of trainable parameters (log-scaled) for fine-tuning a ViT-B/16 pretrained on ImageNet-21k. CaRA demonstrates a performance boost while utilizing only a small fraction of trainable parameters.

parameters in the pre-trained network.

A prominent example of PEFT is Low-Rank Adaptation (LoRA) (Hu et al., 2022), which additively fine-tunes the low-rank intrinsic dimension of the pre-trained transformer. It represents the three-dimensional MHA projection layer in a two-dimensional space $\mathbb{R}^{d_{model} \times h d_k}$ as illustrated in Figure 2. Such fine-tuning in the reduced dimensional space severely limits LoRA's representation ability since, by definition, a two-dimensional representation cannot capture correlations beyond the second axis. While it facilitates simple design and incurs no additional inference costs, a performance gap exists compared to FT (Shuttleworth et al., 2024; Liu et al., 2024).

Tensor calculus (Synge & Schild, 1978; De Lathauwer et al., 2000; Kolda & Bader, 2009; Oseledets, 2011) offers a powerful alternative to defining higher-dimensional rank updates, and recent works in tensor-based low-rank fine-tuning methods (Jie & Deng, 2023; Bershatsky et al., 2024; Yang et al., 2024) have demonstrated significant improvements over LoRA, with fewer trainable parameters.

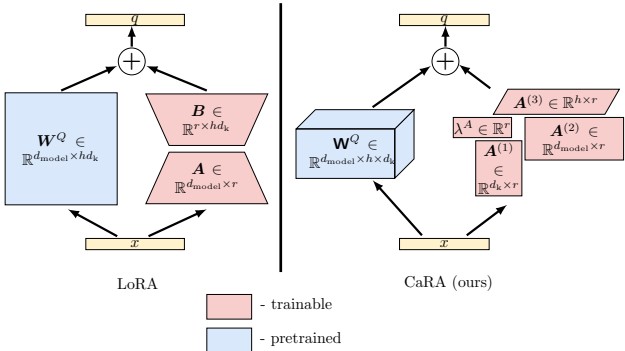

*Figure 2.* Schematic illustration of fine-tuning only a single query projection layer in one transformer block using LoRA on the left and CaRA (ours) on the right. Contrary to LoRA, we represent the low-rank update in CPD form.

For instance, Factor Tuning (FacT) (Jie & Deng, 2023) first tensorise the ViT blocks by stacking all the projection layers into this format $\mathbb{R}^{12l \times d_{model} \times hd_k}$, where 12 represents the number of projection layers in the transformer blocks ($\boldsymbol{W}^Q, \boldsymbol{W}^K, \boldsymbol{W}^V, \boldsymbol{W}^O, \boldsymbol{W}^{\text{up}}$, and $\boldsymbol{W}^{\text{down}}$) and $l$ denotes the number of blocks in the ViT. Later, this work fine-tunes the network using the Tensor-Train and Tucker decomposition format for the low-rank update. Though effective, this formulation still does not account for the high-dimensional nature of projection layers in MHA, as it does not explicitly capture the low-rank update along the head dimension. To this end, we propose a novel method, CaRA, which strategically splits the large tensor into two sub-tensors: one for high-dimensional projection layers in MHA ($\mathbf{W}^{\text{mha}}$) and the other for feed-forward layers ($\mathbf{W}^{\text{ffn}}$).

The tensor $\mathbf{W}^{\text{mha}}$ is prepared by stacking all $\boldsymbol{W}^Q, \boldsymbol{W}^K$, and $\boldsymbol{W}^V$ across $l$ transformer blocks, resulting in a four-dimensional tensor $\mathbb{R}^{3l \times d_{\text{model}} \times h \times d_k}$. Similarly, $\mathbf{W}^{\text{ffn}}$ is created by stacking all other feed-forward layers, particularly $\boldsymbol{W}^O, \boldsymbol{W}^{\text{up}}$, and $\boldsymbol{W}^{\text{down}}$, into a three-dimensional tensor $\mathbb{R}^{9l \times d_{\text{model}} \times d_{\text{model}}}$. This process of unique tensorisation is graphically represented in Figure 3. The proposed split lets CaRA efficiently represent the low-rank update of MHA layers to their maximum capacity; particularly CaRA allows capturing relations across heads. In addition to this tensorisation, we propose using the CPD format for the low-rank update, as depicted in Figure 2. This choice is motivated by the efficiency and ease of fine-tuning (Lebedev et al., 2015; Phan et al., 2020) provided by the CPD format. While tensor decompositions suffer from the curse of dimensionality (Bershatsky et al., 2024), we observe that CPD is less prone to this behaviour than Tensor-Train and Tucker decompositions. These advantages make CPD well-suited for the high-dimensional structure of the proposed tensor.

We conduct extensive experimentation across diverse vision

classification datasets from two benchmarks. Firstly, we evaluate our approach CaRA on the VTAB-1k benchmark and observe that our approach achieves superior performance with only a small fraction of trainable parameters. Figure 1 demonstrates the performance-to-parameter ratio of PEFT methods; it highlights CaRA's efficiency. Secondly, we extend the evaluations to five FGVC benchmark datasets. CaRA achieves competitive performance on these datasets. Thirdly, we analyse the performance of CaRA on a large-scale ViT architecture on various image classification datasets. Finally, we investigate CaRA's behaviour through comprehensive ablations. The first ablation focuses on exploring the effect of rank on CaRA. Notably, we found that CaRA maintains robustness towards rank changes with stable parameter growth. In the second ablation, we study the effect of various tensorisation formulations for $\mathbf{W}^{\text{mha}}$ and $\mathbf{W}^{\text{ffn}}$. Subsequently, through investigating a series of saliency maps, we highlight the learning patterns of CaRA despite its limited parameters. The source code is available at `https://github.com/BonnBytes/CaRA`.

In summary, this paper makes the following contributions:

1. We propose Canonical Rank Adaptation (CaRA), a novel and efficient PEFT method for fine-tuning a Vision Transformer (ViT) that exploits the high-dimensional nature of Multi-Head Attention (MHA).

2. CaRA proposes a novel way of tensorising the transformer into two individual tensors. One is for MHA and the other is for the feed-forward layer. This tensorisation is followed by a novel low-rank update using the CPD form to fine-tune the ViT.

3. CaRA surpasses the performance of various tensor-based methods and LoRA on image classification benchmarks with a significantly smaller fraction of parameters. Through a series of ablations, we demonstrate the robustness of CaRA to varying ranks.

## 2. Related Work

### 2.1. Parameter-Efficient Fine-Tuning (PEFT)

Traditional finetuning involves training the complete ViT network (Dosovitskiy et al., 2021). Recently, PEFT methods have emerged as an attractive approach for fine-tuning large transformer models (Hu et al., 2022; Jia et al., 2022). PEFT provides a significant advantage as it requires only training parts of the network. We broadly arrange the visual PEFT methods into three categories. Firstly, the adapter-based methods introduce additional parameters to the network and fine-tune them. Early works such as Adapter (Houlsby et al., 2019) and Visual Prompt Tuning (VPT) (Jia et al., 2022) established the foundation. Another approach, SSF (Lian et al., 2022), fine-tunes a transformer by learning to scale and shift the individual transformer block features. Recent

approaches such as (Chen et al., 2022; Luo et al., 2023) combine multiple PEFT methods to demonstrate performance improvements. While effective, adapter-based methods add additional inference costs.

The second style uses Neural Architecture Search to optimise the low-rank update. NOAH (Zhang et al., 2024) is an example of performing a search over Adapter, LoRA and VPT for each Transformer block. The final type of PEFT methods is based on LoRA. It was first proposed by Hu et al. (2022) for fine-tuning Large Langauge Models (LLMs), and recently adopted for the vision domain (Zhang et al., 2024; Jie & Deng, 2023). LoRA indirectly optimises low-rank decomposition matrices of dense layers, providing a computational advantage during inference as the low-rank weights are merged into the pre-trained weights. Many variants of LoRA such as Vector-based Random Matrix Adaptation (VeRA) (Kopiczko et al., 2024), NoLA (Koohpayegani et al., 2024), PiSSA (Meng et al., 2024), and Weight Decomposed Low-Rank Adaptation (DoRA) (Liu et al., 2024) are further proposed to reduce the performance gap to FT. SPT-LoRA (He et al., 2023) improves LoRA by identifying sensitive parameters in LoRA's update. A matrix-based representation for the low-rank updates, such as LoRA and its variants, has demonstrated significant potential in fine-tuning but is limited in expressive power by design. While tensor-based representations can capture multi-dimensional correlation, they remain under-explored mainly in the context of PEFT methods. Bershatsky et al. (2024) and Yang et al. (2024) proposed the use of tensor representations for fine-tuning LLMs, and FacT (Jie & Deng, 2023) uses tensor-based methods for vision classification tasks.

The above-mentioned previous works are focused on Tucker and Tensor-Train representations of low-rank updates. An alternative method to represent a tensor low-rank update is the Canonical-Polyadic Decomposition (CPD) (Hitchcock, 1927; 1928). CPD has been demonstrated to be a stable (Phan et al., 2020) and effective method for training convolutional neural networks (Lebedev et al., 2015; Veeramacheneni et al., 2022). To the best of our knowledge, CPD has not been explored as a PEFT method for fine-tuning a ViT. This gap, combined with the limitations of matrix-based representations for fine-tuning high-dimensional MHA layers, motivates us to investigate an effective and novel low-rank update mechanism using CPD.

# 3. Methodology

In this section, we start with a brief introduction to Canonical-Polyadic Decomposition (CPD) (Kolda & Bader, 2009) and LoRA (Hu et al., 2022). Then, we introduce CaRA, particularly the process of tensorisation followed by the low-rank update format. Additionally, we derive gradients for our CaRA formulation.

## 3.1. Preliminaries

### 3.1.1. CANONICAL-POLYADIC DECOMPOSITION (CPD)

Canonical-Polyadic Decomposition (CPD) was first proposed by Hitchcock (1927; 1928) as an idea of decomposing a tensor as a finite sum of rank-one tensors. Historically, it was rediscovered with names such as CANDECOMP-PARAFAC, PARAFAC, CAND or Polyadic tensor form (Kolda & Bader, 2009). Following the notation from (Kolda & Bader, 2009), the CPD of a fourth-order tensor $\mathsf{T} \in \mathbb{R}^{I \times J \times K \times L}$ is mathematically expressed as an outer product of rank-1 vectors

$$\mathsf{T} \approx \{\lambda^S; S^{(1)}, S^{(2)}, S^{(3)}, S^{(4)}\}$$
$$\approx \sum_{r=1}^{R} \lambda_r^S \boldsymbol{s}_r^{(1)} \circ \boldsymbol{s}_r^{(2)} \circ \boldsymbol{s}_r^{(3)} \circ \boldsymbol{s}_r^{(4)}, \qquad (1)$$

where $\circ$ denotes the outer product, $R$ is a positive integer defining the tensor rank, $\lambda^S \in \mathbb{R}^R$, $\boldsymbol{S}^{(1)} \in \mathbb{R}^{I \times R}$, $\boldsymbol{S}^{(2)} \in \mathbb{R}^{J \times R}$, $\boldsymbol{S}^{(3)} \in \mathbb{R}^{K \times R}$, $\boldsymbol{S}^{(4)} \in \mathbb{R}^{L \times R}$ are the CP-Factor matrices with $\boldsymbol{s}_r^{(1)} \in \mathbb{R}^{I \times 1}$, $\boldsymbol{s}_r^{(2)} \in \mathbb{R}^{J \times 1}$, $\boldsymbol{s}_r^{(3)} \in \mathbb{R}^{K \times 1}$, and $\boldsymbol{s}_r^{(4)} \in \mathbb{R}^{L \times 1}$ being the corresponding $r^{th}$ rank one vector in the respective factor matrices. We use this formulation in our proposed approach to represent the low-rank update without normalisation, as this would let us freely scale the $\lambda_r^S$ from the factors. CPD represents a tensor $\mathsf{T}$ of $IJKL$ elements with only $R + R(I + J + K + L)$ elements. This low parameter count is one of the reasons that CPD is a suitable candidate for the PEFT method.

### 3.1.2. LOW-RANK ADAPTATION (LoRA)

LoRA (Hu et al., 2022) proposed an efficient fine-tuning of large networks. Contrary to traditional fine-tuning, LoRA updates the weight additively in the low intrinsic dimension. Consider a linear layer with weight $\boldsymbol{W} \in \mathbb{R}^{n \times m}$, the LoRA weight update is then expressed as

$$y = \boldsymbol{W}x + \delta\boldsymbol{W}x$$
$$= \boldsymbol{W}x + \alpha\boldsymbol{B}\boldsymbol{A}^T x, \qquad (2)$$

where $\boldsymbol{W}$ is a frozen weight matrix during training, $\boldsymbol{B} \in \mathbb{R}^{n \times R}$ and $\boldsymbol{A} \in \mathbb{R}^{m \times R}$ are trainable low-rank matrices, and $\alpha$ is the scaling hyper-parameter. Hu et al. (2022) initialised matrix $\boldsymbol{A}$ with random Gaussian initialisation and matrix $\boldsymbol{B}$ to zeros so that the weight update $\delta\boldsymbol{W}$ in Equation 2 is zero at the beginning of the training. This formulation is graphically represented in Figure 2. Typically, rank $R \ll \min(n, m)$, and this low-rank representation requires only training $(n+m)R$ parameters as opposed to $nm$ parameters in the case of full fine-tuning (FT). Interestingly, $\delta\boldsymbol{W}$ is added to $\boldsymbol{W}$ during inference, promoting no additional latency and inference costs compared to FT.

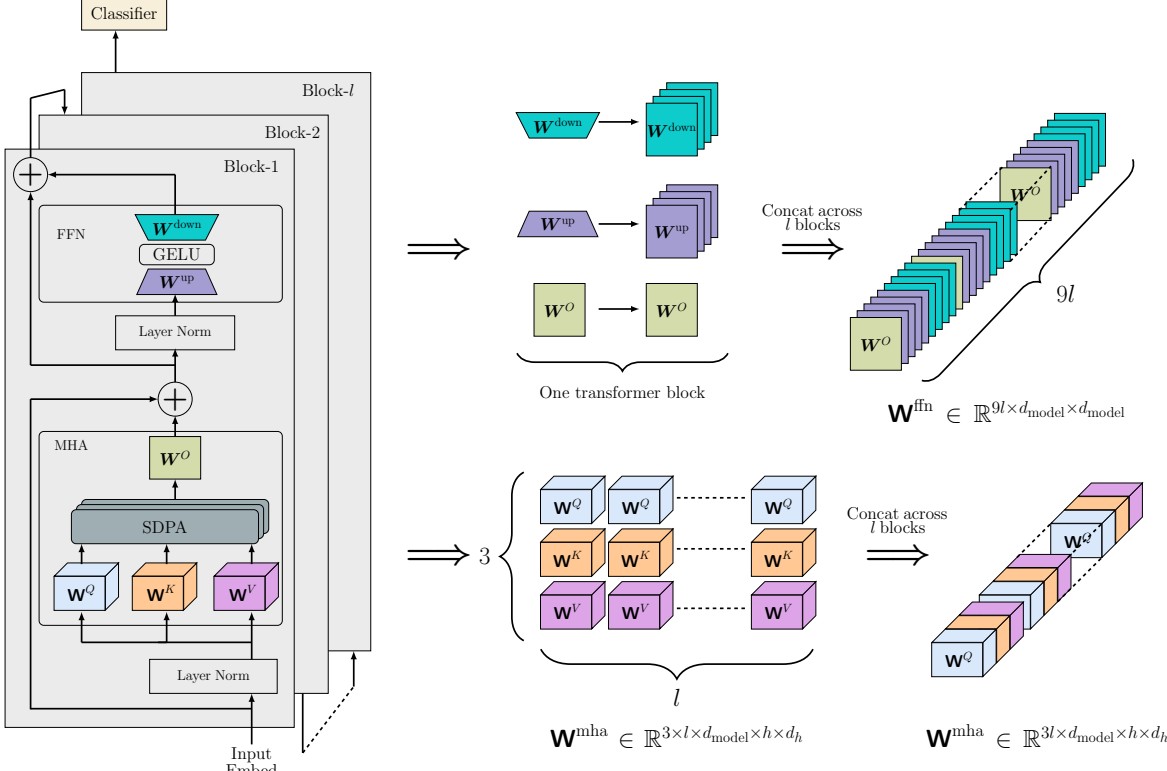

**Figure 3.** Illustration of CaRA's tensorisation process. The figure on the left illustrates the vision transformer with $l$ blocks next to each other. To make a four-dimensional tensor (bottom), we stack all three-dimensional query, key and value projection tensors across $l$ blocks. Similarly, we stack the remaining feed-forward layers to create another tensor (top).

## 3.2. Canonical Rank Adaptation (CaRA)

While LoRA exhibits significant advantages compared to fine-tuning, previous work (Bershatsky et al., 2024; Jie & Deng, 2023) demonstrated that tensor methods for low-rank representation are highly efficient. Given the high-dimensional nature of Multi-Head Attention (MHA), it is evident that utilising tensor representations, especially the Canonical-Polyadic Decomposition (CPD) form, for low-rank updates provides a compact and expressive approach, all while offering a smaller parameter count. This section, presents our novel network-tensorisation approach, followed by our CaRA representation of low-rank updates.

### 3.2.1. TENSORISATION

We propose a novel formulation of tensorising the ViT. It involves creating two tensors: one for MHA's projection layers and a second for the feed-forward layers. This tensorisation will allow us to represent the low-rank update for MHA's projection layers at higher dimensions. Contrary to the existing works, this formulation will particularly allow us to capture any relations across heads in the transformer blocks while being parameter efficient. For the MHA, we stack the query, key and value projection

matrices, i.e., $\boldsymbol{W}^Q \in \mathbb{R}^{d_{\text{model}} \times d_k}$, $\boldsymbol{W}^K \in \mathbb{R}^{d_{\text{model}} \times d_k}$, and $\boldsymbol{W}^V \in \mathbb{R}^{d_{\text{model}} \times d_v}$ for an individual head $i$ in a layer resulting in

$$\mathbf{E}_i = [\boldsymbol{W}_i^Q, \boldsymbol{W}_i^K, \boldsymbol{W}_i^V] \in \mathbb{R}^{3 \times d_{\text{model}} \times d_h} \quad (3)$$

where $d_h$ represents the individual head dimension. In the case of ViT, $d_h = d_v = d_k$ and the enclosing square brackets denote the stacking operation. Furthermore, we stack the resulting tensor $\mathbf{E}_i$ in Equation 3 across all $h$ heads in a specific transformer block $j$

$$\mathbf{L}_j = [\mathbf{E}_1, \mathbf{E}_2, .., \mathbf{E}_i, .., \mathbf{E}_{h-1}, \mathbf{E}_h] \in \mathbb{R}^{3 \times d_{\text{model}} \times h \times d_h}. \quad (4)$$

Finally, we collect all the corresponding $\mathbf{L}_j$ from the $l$ blocks of the transformer and stack them to result in a five-dimensional tensor

$$\mathbf{W}^{\text{mha}} = [\mathbf{L}_1, \mathbf{L}_2, .., \mathbf{L}_j, .., \mathbf{L}_{l-1}, \mathbf{L}_l] \in \mathbb{R}^{3 \times l \times d_{\text{model}} \times h \times d_h}. \quad (5)$$

Empirically, we observe that a combined representation of the first two dimensions, 3 and $l$, in Equation 5 performs better compared to the five-dimensional representation (see Section 5.2). Following this result, we represent the tensor $\mathbf{W}^{\text{mha}}$ as a four-dimensional tensor $\mathbb{R}^{3l \times d_{\text{model}} \times h \times d_h}$.

*Table 1.* Comparison of low-rank updates for various methods such as FacT-TT, FacT-TK, LoRA and CaRA (ours).

| Method | Low-rank update |
|---|---|
| LoRA (Hu et al., 2022) | $\boldsymbol{B}\boldsymbol{A}^T$ |
| FacT-TK (Jie & Deng, 2023) | $\sum_{t_1=1}^{r_1}\sum_{t_2=1}^{r_2}\sum_{t_3=1}^{r_3}\mathsf{G}_{t_1,t_2,t_3}\boldsymbol{P}_{t_1}\boldsymbol{U}_{t_2}\boldsymbol{V}_{t_3}$ |
| FacT-TT (Jie & Deng, 2023) | $\sum_{t_1=1}^{r_1}\sum_{t_2=1}^{r_2}\mathsf{S}_{t_1,t_2}\boldsymbol{U}_{t_1}\boldsymbol{V}_{t_2}$ |
| CaRA (ours) | $\sum_{r=1}^{R}\lambda_r^A\boldsymbol{a}_r^{(1)}\circ\boldsymbol{a}_r^{(2)}\circ\boldsymbol{a}_r^{(3)}\circ\boldsymbol{a}_r^{(4)}$ |

For the second tensor, we consider the linear layers $\boldsymbol{W}^O \in \mathbb{R}^{d_{\text{model}}\times d_{\text{model}}}$ and layers from the position-wise feed-forward network, i.e., $\boldsymbol{W}^{\text{up}} \in \mathbb{R}^{d_{\text{model}}\times d_{\text{ff}}}$ and $\boldsymbol{W}^{\text{down}} \in \mathbb{R}^{d_{\text{ff}}\times d_{\text{model}}}$. Typically, we observe $d_{\text{ff}} = 4d_{\text{model}}$ in the case of ViT, and subsequently, $\boldsymbol{W}^{\text{up}}$ and $\boldsymbol{W}^{\text{down}}$ are represented as three-dimensional tensors with shape $\mathbb{R}^{4\times d_{\text{model}}\times d_{\text{model}}}$. Analogous to the previous tensor, we create a tensor per transformer block $j$ by stacking $\boldsymbol{W}^O$, $\mathsf{W}^{\text{up}}$, and $\mathsf{W}^{\text{down}}$, resulting in

$$\mathsf{F}_j = [\mathsf{W}^O, \mathsf{W}^{\text{up}}, \mathsf{W}^{\text{down}}] \in \mathbb{R}^{9\times d_{\text{model}}\times d_{\text{model}}}, \qquad (6)$$

where $\mathsf{W}^O \in \mathbb{R}^{1\times d_{\text{model}}\times d_{\text{model}}}$ is the tensorised version of linear layer $\boldsymbol{W}^O$ by adding an additional dimension. Finally, we concatenate $\mathsf{F}_j$ from Equation 6 across $l$ blocks of the transformer, resulting in

$$\mathsf{W}^{\text{ffn}} = [\mathsf{F}_1, \mathsf{F}_2, .., \mathsf{F}_j, .., \mathsf{F}_{l-1}, \mathsf{F}_l] \in \mathbb{R}^{9l\times d_{\text{model}}\times d_{\text{model}}}. \quad (7)$$

This final step completes the tensorisation of the transformer architecture, and this process is graphically presented in Figure 3. Unlike prior work, where all the blocks are aggregated into one single tensor, we decouple the MHA and feed-forward networks into separate tensors. This split particularly enables CaRA to represent MHA as a four-dimensional tensor, allowing a richer representation compared to traditional three-dimensional representations (Bershatsky et al., 2024; Jie & Deng, 2023). Additionally, this allows us to represent the low-rank updates along the head basis separately, as depicted in Figure 2. Empirically, we demonstrate in Section 5.2 that this split in an MHA tensor and FFN tensor increases the accuracy compared to a single tensorisation.

### 3.2.2. CaRA LOW-RANK REPRESENTATION

With the tensorised network in place, we propose a novel low-rank update representation leveraging the CPD format for fine-tuning the ViT. Using the CPD format for low-rank updates, as depicted in Figure 1, notably offers benefits in terms of parameter efficiency. To represent a low-rank update for the tensor $\mathsf{W}^{\text{mha}}$, we define the update in CPD format to be $\delta\mathsf{W}^{\text{mha}}$ as

$$\delta\mathsf{W}^{\text{mha}} = \{\lambda^A; \boldsymbol{A}^{(1)}, \boldsymbol{A}^{(2)}, \boldsymbol{A}^{(3)}, \boldsymbol{A}^{(4)}\}$$
$$= \sum_{r=1}^{R}\lambda_r^A\boldsymbol{a}_r^{(1)}\circ\boldsymbol{a}_r^{(2)}\circ\boldsymbol{a}_r^{(3)}\circ\boldsymbol{a}_r^{(4)}, \qquad (8)$$

where $\{\}$ represent the set of CPD factors in matrix form, $\boldsymbol{A}^{(1)} \in \mathbb{R}^{3l\times R}$, $\boldsymbol{A}^{(2)} \in \mathbb{R}^{d_{\text{model}}\times R}$, $\boldsymbol{A}^{(3)} \in \mathbb{R}^{h\times R}$, and $\boldsymbol{A}^{(4)} \in \mathbb{R}^{d_h\times R}$ are the four-factor matrices, $\lambda^A \in \mathbb{R}^R$ is the learned scaling factor, $a_r^{(n)}$ represent the $r^{th}$ rank column vector in $n^{th}$ factor, and $\circ$ represents the outer product. The fine-tuned weight update in Equation 8, not to be confused with the gradient descent update, is expressed as

$$\mathsf{W}^{\text{mha}}+\alpha\delta\mathsf{W}^{\text{mha}} = \mathsf{W}^{\text{mha}}+\alpha\sum_{r=1}^{R}\lambda_r^A\boldsymbol{a}_r^{(1)}\circ\boldsymbol{a}_r^{(2)}\circ\boldsymbol{a}_r^{(3)}\circ\boldsymbol{a}_r^{(4)}.$$
$$(9)$$

Similarly, for the three-dimensional tensor $\mathsf{W}^{\text{ffn}}$, we get only three factors in CPD format

$$\delta\mathsf{W}^{\text{ffn}} = \{\lambda^B; \boldsymbol{B}^{(1)}, \boldsymbol{B}^{(2)}, \boldsymbol{B}^{(3)}\}$$
$$= \sum_{r=1}^{R}\lambda_r^B\boldsymbol{b}_r^{(1)}\circ\boldsymbol{b}_r^{(2)}\circ\boldsymbol{b}_r^{(3)}, \qquad (10)$$

where $\boldsymbol{B}^{(1)} \in \mathbb{R}^{9l\times R}$, $\boldsymbol{B}^{(2)} \in \mathbb{R}^{d_{\text{model}}\times R}$, $\boldsymbol{B}^{(3)} \in \mathbb{R}^{d_{\text{model}}\times R}$ are the factor matrices, $\lambda^B \in \mathbb{R}^R$ is the learned scaling factor. Finally, we write the fine-tuned weight update for $\mathsf{W}^{\text{ffn}}$ as

$$\mathsf{W}^{\text{ffn}} + \alpha\delta\mathsf{W}^{\text{ffn}} = \mathsf{W}^{\text{ffn}} + \alpha\sum_{r=1}^{R}\lambda_r^B\boldsymbol{b}_r^{(1)}\circ\boldsymbol{b}_r^{(2)}\circ\boldsymbol{b}_r^{(3)}.$$
$$(11)$$

Figure 2 graphically illustrates our weight update for one of the query projections in a specific transformer block. Table 1 presents a comparative summary of various tensor-based low-rank updates.

### 3.3. CaRA Gradients

We fine-tune the proposed formulation using PyTorch autograd (Paszke et al., 2017) and the gradient descent technique. This section provides the gradient for our low-rank update concerning individual rank vectors in $\delta\mathsf{W}^{\text{mha}}$ and $\delta\mathsf{W}^{\text{ffn}}$ as in Equations 9 and 11, respectively. Consider a loss function $L$; the gradients for the MHA update are defined as

$$\nabla_{\lambda_r^A} L = \boldsymbol{a}_r^{(1)}\circ\boldsymbol{a}_r^{(2)}\circ\boldsymbol{a}_r^{(3)}\circ\boldsymbol{a}_r^{(4)}, \qquad (12)$$
$$\nabla_{\boldsymbol{a}_r^{(1)}} L = \boldsymbol{I}^{A1}\circ\boldsymbol{a}_r^{(2)}\circ\boldsymbol{a}_r^{(3)}\circ\boldsymbol{a}_r^{(4)}, \qquad (13)$$
$$\nabla_{\boldsymbol{a}_r^{(2)}} L = \boldsymbol{I}^{A2}\circ\boldsymbol{a}_r^{(1)}\circ\boldsymbol{a}_r^{(3)}\circ\boldsymbol{a}_r^{(4)}, \qquad (14)$$
$$\nabla_{\boldsymbol{a}_r^{(3)}} L = \boldsymbol{I}^{A3}\circ\boldsymbol{a}_r^{(1)}\circ\boldsymbol{a}_r^{(2)}\circ\boldsymbol{a}_r^{(4)}, \qquad (15)$$
$$\nabla_{\boldsymbol{a}_r^{(4)}} L = \boldsymbol{I}^{A4}\circ\boldsymbol{a}_r^{(1)}\circ\boldsymbol{a}_r^{(2)}\circ\boldsymbol{a}_r^{(3)}, \qquad (16)$$

where $\boldsymbol{I}^{A1} \in \mathbb{R}^{3l\times 3l}$, $\boldsymbol{I}^{A2} \in \mathbb{R}^{d_{\text{model}}\times d_{\text{model}}}$, $\boldsymbol{I}^{A3} \in \mathbb{R}^{h\times h}$, and $\boldsymbol{I}^{A4} \in \mathbb{R}^{d_h\times d_h}$ are identity matrices. Similarly, gradi-

*Table 2.* VTAB-1k evaluation results with VIT-B/16 backbone on a wide range of 19 datasets. Our method results are highlighted in grey. The number of parameters is averaged over group-wise mean values. We indicate both group-wise and overall mean accuracy. We present the best result in bold and the second best as underlined. Standard deviation per dataset over 10 runs is presented in Table 8 of the Appendix.

| | #param (M) | Natural | | | | | | | Specialized | | | | Structured | | | | | | | | Group Mean | Overall Mean |
| | | Cifar100 | Caltech101 | DTD | Flower102 | Pets | SVHN | Sun397 | Camelyon | EuroSAT | Resisc45 | Retinopathy | Clevr-Count | Clevr-Dist | DMLab | KITTI-Dist | dSpr-Loc | dSpr-Ori | sNORB-Azim | sNORB-Ele | | |
|---|---|---|---|---|---|---|---|---|---|---|---|---|---|---|---|---|---|---|---|---|---|---|
| **Traditional Fine-Tuning** | | | | | | | | | | | | | | | | | | | | | | |
| Linear | - | 63.4 | 85.0 | 63.2 | 97.0 | 86.3 | 36.6 | 51.0 | 78.5 | 87.5 | 68.6 | 74.0 | 34.3 | 30.6 | 33.2 | 55.4 | 12.5 | 20.0 | 9.6 | 19.2 | 57.64 | 52.94 |
| FT | 85.8 | 68.9 | 87.7 | 64.3 | 97.2 | 86.9 | 87.4 | 38.8 | 79.7 | 95.7 | 84.2 | 73.9 | 56.3 | 58.6 | 41.7 | 65.5 | 57.5 | 46.7 | 25.7 | 29.1 | 68.96 | 65.57 |
| **PEFT methods** | | | | | | | | | | | | | | | | | | | | | | |
| *Adapter based* | | | | | | | | | | | | | | | | | | | | | | |
| Adapter-256 | 0.27 | 74.1 | 86.1 | 63.2 | 97.7 | 87.0 | 34.6 | 50.8 | 76.3 | 88.0 | 73.1 | 70.5 | 45.7 | 37.4 | 31.2 | 53.2 | 30.3 | 25.4 | 13.8 | 22.1 | 59.95 | 55.82 |
| VPT-Shallow | 0.06 | 77.7 | 86.9 | 62.6 | 97.5 | 87.3 | 74.5 | 51.2 | 78.2 | 92.0 | 75.6 | 72.9 | 50.5 | 58.6 | 40.5 | 67.1 | 68.7 | 36.1 | 20.2 | 34.1 | 67.82 | 64.85 |
| VPT-Deep | 0.53 | 78.8 | 90.8 | 65.8 | 98.0 | 88.3 | 78.1 | 49.6 | 81.8 | 96.1 | 83.4 | 68.4 | 68.5 | 60.0 | 46.5 | 72.8 | 73.6 | 47.9 | 32.9 | 37.8 | 71.96 | 69.43 |
| AdaptFormer | 0.16 | 70.8 | 91.2 | 70.5 | 99.1 | 90.9 | 86.6 | 54.8 | 83.0 | 95.8 | 84.4 | 76.3 | 81.9 | 64.3 | 49.3 | 80.3 | 76.3 | 45.7 | 31.7 | 41.1 | 74.75 | 72.32 |
| SSF | 0.24 | 69.0 | 92.6 | 75.1 | 99.4 | 91.8 | 90.2 | 52.9 | 87.4 | 95.9 | 87.4 | 75.5 | 75.9 | 62.3 | 53.3 | 80.6 | 77.3 | 54.9 | 29.5 | 37.9 | 75.69 | 73.10 |
| RepAdapter | 0.22 | 72.4 | 91.6 | 71.0 | 99.2 | 91.4 | 90.7 | 55.1 | 85.3 | 95.9 | 84.6 | 75.9 | 82.3 | 68.0 | 50.4 | 79.9 | 80.4 | 49.2 | 38.6 | 41.0 | 76.09 | 73.84 |
| *NAS based* | | | | | | | | | | | | | | | | | | | | | | |
| NOAH | 0.361 | 69.6 | 92.7 | 70.2 | 99.1 | 90.4 | 86.1 | 53.7 | 84.4 | 95.4 | 83.9 | 75.8 | 82.8 | 68.9 | 49.9 | 81.7 | 81.8 | 48.3 | 32.8 | 44.2 | 75.5 | 73.25 |
| *LoRA based* | | | | | | | | | | | | | | | | | | | | | | |
| LoRA | 0.29 | 67.1 | 91.4 | 69.4 | 98.8 | 90.4 | 85.3 | 54.0 | 84.9 | 95.3 | 84.4 | 73.6 | 82.9 | 69.2 | 49.8 | 78.5 | 75.7 | 47.1 | 31.0 | 44.0 | 74.60 | 72.25 |
| FacT-TT | 0.04 | 71.3 | 89.6 | 70.7 | 98.9 | 91.0 | 87.8 | 54.6 | 85.2 | 95.5 | 83.4 | 75.7 | 82.0 | 69.0 | 49.8 | 80.0 | 79.2 | 48.4 | 34.2 | 41.4 | 75.34 | 73.04 |
| FacT-TK | 0.07 | 70.6 | 90.6 | 70.8 | 99.1 | 90.7 | 88.6 | 54.1 | 84.8 | 96.2 | 84.5 | 75.7 | 82.6 | 68.2 | 49.8 | 80.7 | 80.8 | 47.4 | 33.2 | 43.0 | 75.56 | 73.23 |
| SPT-LoRA | 0.54 | 73.5 | 93.3 | 72.5 | 99.3 | 91.5 | 87.9 | 55.5 | 85.7 | 96.2 | 85.9 | 75.9 | 84.4 | 67.6 | 52.5 | 82.0 | 81.0 | 51.1 | 30.2 | 41.3 | 76.37 | 74.07 |
| CaRA (ours) | 0.06 | 71.3 | 91.9 | 71.8 | 99.3 | 91.4 | 90.5 | 54.7 | 86.2 | 96.4 | 86.0 | 75.4 | 83.8 | 69.4 | 51.4 | 81.7 | 80.8 | 47.4 | 35.3 | 44.0 | 76.46 | 74.14 |

ents for the feed-forward low-rank updates are given by

$$\nabla_{\lambda_r^B} L = \boldsymbol{b}_r^{(1)} \circ \boldsymbol{b}_r^{(2)} \circ \boldsymbol{b}_r^{(3)}, \tag{17}$$

$$\nabla_{\boldsymbol{b}_r^{(1)}} L = \boldsymbol{I}^{B1} \circ \boldsymbol{b}_r^{(2)} \circ \boldsymbol{b}_r^{(3)}, \tag{18}$$

$$\nabla_{\boldsymbol{b}_r^{(2)}} L = \boldsymbol{I}^{B2} \circ \boldsymbol{b}_r^{(1)} \circ \boldsymbol{b}_r^{(3)}, \tag{19}$$

$$\nabla_{\boldsymbol{b}_r^{(3)}} L = \boldsymbol{I}^{B3} \circ \boldsymbol{b}_r^{(1)} \circ \boldsymbol{b}_r^{(2)}, \tag{20}$$

where $\boldsymbol{I}^{B1} \in \mathbb{R}^{9l \times 9l}$, $\boldsymbol{I}^{B2} \in \mathbb{R}^{d_{\text{model}} \times d_{\text{model}}}$, and $\boldsymbol{I}^{B3} \in \mathbb{R}^{d_{\text{model}} \times d_{\text{model}}}$ are also identity matrices. The proof for the gradients is provided in Appendix A.

Considering a projection layer in MHA, LoRA needs gradients for $(d_{\text{model}} + h \cdot d_h)R$ parameters, whereas CaRA needs to update only $(d_{\text{model}} + h + d_h)R$ parameters. Mathematically, CaRA's low-rank update has fewer degrees of freedom, that is, fewer parameters. This constraint forces the network to learn distinct features, and for this purpose, we study the saliency maps of CaRA in Section 5.3.

### 3.4. CaRA Initialisation

To initialise the above-mentioned factor matrices, we experimented with various initialisations presented in Appendix Section B to determine the most effective initialisation. For matrices $\boldsymbol{A}^{(1)}$ and $\boldsymbol{B}^{(1)}$, we use random normal initialisation (Glorot & Bengio, 2010). Matrices $\boldsymbol{A}^{(2)}$ and $\boldsymbol{B}^{(2)}$ are initialised to zeros to ensure the corresponding $\delta\mathbf{W}$ is initially zero. The remaining factor matrices $\boldsymbol{A}^{(3)}$, $\boldsymbol{A}^{(4)}$, and $\boldsymbol{B}^{(3)}$ are initialised as orthogonal matrices (Saxe et al., 2014). More information about $\lambda$'s initialisation is given in

Appendix B. Similar to LoRA, the weight updates $\delta\mathbf{W}^{\text{mha}}$ and $\delta\mathbf{W}^{\text{ffn}}$ are added to pretrained weights $\mathbf{W}^{\text{mha}}$ and $\mathbf{W}^{\text{ffn}}$ during inference. This design choice incurs no additional latency and inference costs.

## 4. Experiments

We evaluate the performance of CaRA on various vision classification datasets and further perform extensive ablations to demonstrate its effectiveness and explain our design.

### 4.1. Visual Task Adaptation Benchmark (VTAB)

To evaluate the performance of CaRA, we follow the experimental setup from (Jia et al., 2022) and benchmark on all VTAB-1k datasets (Zhai et al., 2019). We use the ImageNet-21k pretrained ViT-B/16 (Dosovitskiy et al., 2021) backbone. **Datasets, Metrics and Hyperparameters.** We use VTAB-1k, a transfer learning benchmark consisting of 19 diverse vision datasets, arranged into three groups: Natural, Structured and Specialized. The datasets in this benchmark, for example, include CIFAR-100, Resisc45 and KITTI. CaRA is trained on a subset of 1000 samples with an 80-20 split for training and validation, while the original test set is used for evaluation. We report Top-1 accuracy for each dataset in the benchmark for evaluation. We present the hyperparameters, such as rank, in Table 8 of the Appendix, and additional information on the datasets is

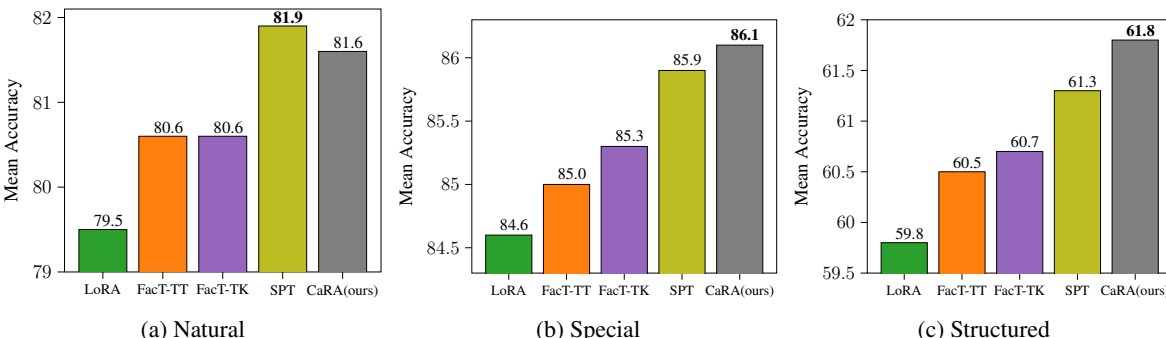

*Figure 4.* Group wise average results on VTAB-1k. While CaRA performs well in Structured and Specialized domains, its performance in Natural datasets is competitive with SPT-LoRA, which requires 6.75 times more trainable parameters than CaRA.

further provided in Section C.2 of the Appendix.

**Results.** Table 2 presents the results of fine-tuning the ViT-B on 19 VTAB-1k datasets. Fine-tuning with CaRA requires only $0.06M$ parameters, which are averaged across all the datasets. CaRA is comparable in parameters to VPT-shallow and marginally higher than the Tensor-Train representation. This marginal parameter increase relative to FacT-TT arises from using a slightly lower rank in the case of FacT-TT. Figure 5 explores the performance of CaRA across varying ranks, demonstrating similar or higher performance compared to FacT-TT at lower ranks.

Table 2 establishes that CaRA outperforms existing visual PEFT methods with only a portion of their trainable parameters. In particular, CaRA achieves state-of-the-art (SOTA) performance on two datasets. Numbers are presented in bold. Interestingly, we observe that CaRA significantly narrows the gap to SOTA methods on eight datasets. We underlined these numbers in the table. Figure 4 depicts the group-wise average accuracy of VTAB-1k datasets. CaRA demonstrates a significant improvement in comparison to existing methods in the case of Special and Structured datasets, whereas for the Natural group, CaRA performs similarly to SPT-LoRA but surpasses LoRA and traditional tensor-based methods.

Overall, in terms of mean accuracy, CaRA outperforms traditional LoRA by $2\%$ and demonstrates higher performance than existing tensor methods by $1\%$. Compared to SPT-LoRA, which additionally identifies and trains sensitive parameters in the LoRA formulation, it achieves similar performance to CaRA at the expense of requiring many more trainable parameters. This result highlights that CaRA further improves the fine-tuning capability of the ViT-B with a significantly smaller fraction of trainable parameters. Moreover, as depicted in Figure 4, CaRA performs best on the challenging datasets of the Special and Structured group, which mainly cover domains that are highly dissimilar to ImageNet.

*Table 3.* Evaluation results on the FGVC benchmark with ViT-B/16 backbone pretrained on ImageNet-21k. We present the best result in bold and second best as underlined. Our method is highlighted in grey.

| | #Params (M) | CUB-200-2011 | NABirds | Flowers | Stanford Dogs | Stanford Cars | Mean |
|---|---|---|---|---|---|---|---|
| **Traditional Fine-Tuning** | | | | | | | |
| Linear | - | 85.3 | 75.9 | 97.9 | 86.2 | 51.3 | 79.32 |
| FT | 85.8 | 87.3 | 82.7 | 98.8 | 89.4 | 84.5 | 88.54 |
| **PEFT methods** | | | | | | | |
| VPT-Shallow | **0.06** | 86.7 | 78.8 | 98.4 | 90.7 | 68.7 | 84.66 |
| LoRA | 0.29 | 85.6 | 79.8 | 98.9 | 87.6 | 72.0 | 84.78 |
| AdaptFormer | 0.16 | 84.7 | 75.2 | 97.9 | 84.7 | 83.1 | 85.12 |
| Adapter | 0.16 | 87.2 | 84.3 | 98.5 | 89.6 | 68.4 | 85.60 |
| VPT-Deep | 0.53 | 88.5 | 84.2 | 99.0 | 90.2 | 83.6 | 89.10 |
| SPT-LoRA | 0.54 | **88.6** | 83.4 | **99.5** | 91.4 | **87.3** | 90.04 |
| CaRA (ours) | 0.08 | **88.6** | **86.5** | 99.4 | **91.5** | 86.2 | **90.46** |

### 4.2. Fine-Grained Visual Categorization (FGVC)

Having shown CaRA performance in fine-tuning VTAB-1k, we further assess the performance on FGVC datasets, using the same ImageNet-21k pretrained ViT-B/16 backbone.

**Datasets, Metrics and Hyperparameters.** FGVC is a collection of five large datasets: CUB-200-2011, NABirds, Oxford Flowers, Stanford Dogs and Stanford Cats. More about the dataset statistics, including the training and validation set sizes, are given in Section C.3 of the Appendix. Unlike VTAB-1k, FGVC utilises the whole training set. We report Top-1 accuracy for evaluation. CaRA is trained with rank 32 across all the datasets. Section C.3 of the Appendix provides more details about hyperparameters.

**Results.** Table 3 shows the performance of CaRA on FGVC. As observed earlier, CaRA achieves competitive performance with only a small fraction of trainable parameters: $0.08M$. Compared to LoRA, VPT-Shallow and other methods, we showcase a significant improvement and reach performance levels comparable to those of SPT-LoRA. CaRA achieves SOTA performance on three out of five datasets, such as CUB-200-2011, NABirds, and Stanford Dogs. In the case of the other datasets, Stanford Cars and Oxford-

*Table 4.* Evaluation results on the ViT-Large backbone pretrained on ImageNet-21k. We present the best result in bold and the second best as underlined. Our method is highlighted in grey.

| | #Params (M) | CIFAR100 | Food101 | Flowers102 | Resisc45 | Mean |
|---|---|---|---|---|---|---|
| **Traditional Fine-Tuning** | | | | | | |
| Linear | - | 79.4 | 76.5 | 98.9 | 67.8 | 80.65 |
| Full | 303.3 | 86.8 | 78.7 | 98.8 | 79.0 | 85.83 |
| **PEFT methods** | | | | | | |
| LoRA | 0.786 | 87.0 | 79.5 | 99.1 | 78.3 | 85.98 |
| VeRA | **0.061** | 87.5 | 79.2 | 99.2 | 78.6 | 86.13 |
| PiSSA | 0.786 | 87.1 | 79.6 | **99.7** | 78.6 | 86.24 |
| DoRA | 0.860 | 87.9 | 81.2 | 99.6 | 80.3 | 87.60 |
| CaRA (ours) | 0.076 | **89.4** | **83.7** | 99.6 | **82.4** | **88.77** |

Flowers, CaRA performs on par with SPT-LoRA and VPT-Deep. Overall, these results, combined with the previous experiments, demonstrate that our proposed tensor-based CaRA significantly outperforms traditional LoRA, adapter and other tensor-based methods in vision classification tasks while utilising only a small fraction of trainable parameters.

### 4.3. Fine-tuning ViT-L

To evaluate the performance of CaRA on a larger vision transformer, we fine-tune ViT-Large pretrained on ImageNet-21k across four datasets.

**Datasets, Metrics and Hyperparameters.** Following Kopiczko et al. (2024), we fine-tune on CIFAR100, Food101, Flowers102, and Resisc45 using 10 randomly sampled training examples per class. Evaluation is performed on the CIFAR100, Food101, and Flowers102 test sets, and on the remaining samples for Resisc45. Further implementation and hyperparameter details are provided in Section C.4.

**Results.** Table 4 reports the performance of CaRA across all four datasets. As observed in other experiments, CaRA achieves SOTA results using only $\approx 10\%$ of the trainable parameters compared to LoRA. Regarding parameter count, CaRA is comparable to VeRA, yet it outperforms VeRA by a margin of 2%. Despite the significantly smaller fraction of trainable parameters, these results showcase CaRA's scaling capability and effectiveness in a few-shot learning regime.

## 5. Ablation Study

For a better understanding of our proposed CaRA, we perform comprehensive ablations. We study CaRA's robustness to rank, followed by the effect of dimensionality in tensorisation. Subsequently, we investigate the saliency maps and analyse CaRA's computational complexity. All ablations are performed on VTAB-1k's Special group.

### 5.1. Robustness to Rank

In this subsection, we explore the effect of rank on the performance of CaRA. Accordingly, we trained CaRA with

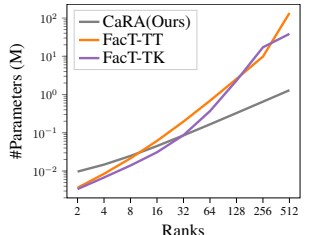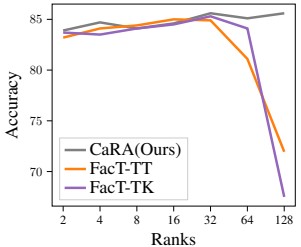

*Figure 5.* CaRA's robustness to rank. The left figure depicts the parameter growth in various tensor formulations with varying rank. The figure on the right illustrates the mean accuracy of various tensor formulations evaluated on the Special group in VTAB-1k with varying ranks. Best viewed in colour.

varying ranks to the powers of two. For comparison, we trained the Tucker and Tensor-Train low-rank representations proposed in FacT (Jie & Deng, 2023). Figure 5 on the left depicts the parameter growth with rank increase. We observe that the number of parameters for FacT-TT and FacT-TK grow faster than CaRA. This phenomenon is due to tensors in Tensor-Train and Tucker formulations for low-rank updates. In contrast, the CaRA representation only contains matrices, as presented in Table 1. Figure 5 on the right demonstrates the evaluation of FacT-TT, FacT-TK, and CaRA with varying ranks. The effect of exponential parameter growth negatively impacts the performance of the Tensor-Train and Tucker methods, whereas CaRA's performance slightly increases with an increase in rank. This result demonstrates the robustness of CaRA to rank and its efficient use of parameters.

### 5.2. Dimension Ablation

As discussed in Section 3.2.1, our formulation allows the low-rank update of $\mathbf{W}^{\text{mha}}$ to be represented at a maximum of five dimensions. In this subsection, we investigate the effect of dimensionality on the performance of various low-rank update dimensions with the Special group of datasets in VTAB-1k. Table 5 demonstrates the effect of dimensionality. Despite slightly smaller trainable parameters in a five-dimensional representation, it is less effective than the four-dimensional format. We hypothesise this effect to the merged QKV projection implementation in the pretrained backbone (Dosovitskiy et al., 2021), contrary to the individual Q, K, and V projections as described in (Vaswani et al., 2017). While the three-dimensional representation achieves a similar accuracy as the proposed presentation, this improvement comes at the cost of a rise in trainable parameters. Moreover, we trained our model with a combined tensor of $\mathbf{W}^{\text{mha}}$ and $\mathbf{W}^{\text{ffn}}$, as shown in the last row of Table 5. We observe that the two separate factorisations achieve higher accuracy than the combined representation. Altogether, from this ablation, we learn that the four-dimensional representation yields better performance

*Table 5.* Dimension ablation experiment evaluated on Special group of VTAB-1k. All experiments are conducted with the rank of 32. Dimensions for $\mathbf{W}^{\text{ffn}}$ is $9l \times d_{\text{model}} \times d_{\text{model}}$ except last line where it is merged into the MHA tensor.

| Tensors | MHA-Dimension | #Params (M) | Accuracy |
|---|---|---|---|
| $\mathbf{W}^{\text{mha}}$-5D, $\mathbf{W}^{\text{ffn}}$ | $3 \times l \times d_{\text{model}} \times h \times d_h$ | 0.08476 | 84.682 |
| $\mathbf{W}^{\text{mha}}$-3D, $\mathbf{W}^{\text{ffn}}$ | $3l \times d_{\text{model}} \times hd_h$ | 0.10758 | 85.061 |
| $\mathbf{W}^{\text{mha}}$-4D, $\mathbf{W}^{\text{ffn}}$ (CaRA) | $3l \times d_{\text{model}} \times h \times d_h$ | 0.08544 | **85.134** |
| $\mathbf{W}^{\text{mha}}$-3D + $\mathbf{W}^{\text{ffn}}$ | $12l \times d_{\text{model}} \times d_{\text{model}}$ | **0.05840** | 83.914 |

*Table 6.* CaRA's computational complexity with walltime measured in seconds and VRAM occupied in gigabytes.

| Method | LoRA | DoRA | FacT-TT | FacT-TK | CaRA (ours) |
|---|---|---|---|---|---|
| Walltime ($\downarrow$) | **165.756** | 204.076 | 178.283 | 180.578 | 206.554 |
| VRAM ($\downarrow$) | **20.108** | 28.065 | 20.246 | 20.244 | 21.374 |

and lower trainable parameters. Additionally, the disentangled low-rank update for MHA and FFN demonstrates a significant improvement in accuracy.

### 5.3. Interpretability

This subsection attempts to understand CaRA's fine-tuning capability. Subsequently, we examine the saliency maps generated from Integrated Gradients (Sundarararajan et al., 2017).

**Dataset and Model.** We use the FGVC-Aircraft dataset (Maji et al., 2013), which specialises in identifying the various aircraft families. This dataset is ideal for saliency studies because aircraft are part of the ImageNet dataset, but the ImageNet pre-trained model cannot differentiate between aircraft families. Thus, FGVC-Aircraft constitutes an ideal dataset for fine-tuning. The Appendix Section E presents more details on the dataset.

Figure 6 in the Appendix depicts the saliency maps of a ViT-B fine-tuned with ranks 32 and 16 for various aircraft. Saliency maps from rank 32 show a concentrated gradient over specific aircraft parts, while for rank 16, they are more spread out. This difference is likely attributed to significantly lower parameters in the rank-16 model, resulting in a broader focus. Additionally, as humans, we rely on distinct aircraft features to identify them, like the cockpit bump on the B-747 or the third aft engine on the L-1011. The network also focuses on these features. This highlights CaRA's ability to learn distinct and discriminative features, providing effective fine-tuning with a limited set of trainable parameters.

### 5.4. Computational Analysis

In Table 6, we report the training time and GPU memory usage during training for various PEFT methods for fine-tuning ViT-L on CIFAR100. Note that these methods do not incur additional overhead during inference, as the low-rank weights are merged into the corresponding pre-trained weights. We observe that LoRA is most efficient in terms of both training time and memory. We attribute the efficient training time of LoRA to Compute Unified Device Architecture (CUDA)-optimised matrix multiplications in PyTorch. In contrast, CaRA's code is not optimised, as underlying packages are primarily written in Python. Moreover, CaRA's multi-dimensional nature results in slightly higher CUDA memory. In general, the memory consumption of the methods is very similar. Only DoRA requires much more memory due to the additional weight normalisation. Despite the slightly higher walltime and memory consumption compared to LoRA and FacT, CaRA demonstrates performance improvements in both ViT base and large architectures.

## 6. Discussion

In this work, we have introduced a novel and effective tensorisation of a vision transformer, explicitly addressing the high-dimensional nature of Multi-Head Attention (MHA) during fine-tuning. Our proposed Canonical Rank Adaptation (CaRA) exploits this tensorisation alongside the efficiency of the Canonical-Polyadic Decomposition (CPD) form. CaRA consistently outperforms tensor-based and matrix-based low-rank update methods across a diverse range of vision datasets, demonstrating its robustness. Notably, CaRA achieves these results with a significantly smaller number of parameters, fully establishing its place in the visual Parameter-Efficient Fine-Tuning (PEFT) family. Furthermore, the CaRA factors can be merged after fine-tuning into the pre-trained weights, ensuring no additional overhead for inference.

Despite its promising results, our approach has some limitations. The training time of our tensor-based implementation is higher compared to the matrix-based approach LoRA. A hardware-optimised tensor decomposition, such as (Yang et al., 2022; Kao et al., 2022), might provide a more efficient implementation. As part of future work, a weight-normalised form of CaRA could be investigated. The benefit of adding weight normalisation to LoRA has been shown in DoRA (Liu et al., 2024) and a similar form exists for CPD (Kolda & Bader, 2009).

### Impact Statement

Parameter-Efficient Fine-Tuning (PEFT) methods such as Canonical Rank Adaptation (CaRA) reduce the computational cost, and thus the environmental footprint, for fine-tuning very large transformer architectures, including vision, language and multi-modal models.

## Acknowledgements

This research was supported by the Federal Ministry of Education and Research (BMBF) under grant no. 01IS22094A WEST-AI and 6DHBK1022 BNTrAInee, and the Deutsche Forschungsgemeinschaft (DFG, German Research Foundation) under Germany's Excellence Strategy - EXC 2070–390732324 (PhenoRob) and project no. 459420781 (for-5361). Prof. Kuehne is supported by the BMBF project STCL-01IS22067. We thank our anonymous reviewers for helping us improve this publication. The authors gratefully acknowledge the HPC team at the University of Bonn and RWTH Aachen for compute time on the Bender system and CLAIX-2023 system under project rwth1629, respectively. The authors gratefully acknowledge the Gauss Centre for Supercomputing e.V. (www.gauss-centre.eu) for funding this project by providing computing time through the John von Neumann Institute for Computing (NIC) on the GCS Supercomputer JUWELS at Jülich Supercomputing Centre (JSC). They also thank Zahra Ganji for reimplementing VeRA's vision baseline. The authors are solely responsible for the content of this publication.

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

## Acronyms

**CaRA** Canonical Rank Adaptation

**CPD** Canonical-Polyadic Decomposition

**CUDA** Compute Unified Device Architecture

**DoRA** Weight Decomposed Low-Rank Adaptation

**FacT** Factor Tuning

**FGVC** Fine-Grained Visual Categorization

**FT** full fine-tuning

**GLUE** General Language Understanding Evaluation

**LLMs** Large Langauge Models

**LoRA** Low-Rank Adaptation

**MHA** Multi-Head Attention

**NLU** Natural Language Understanding

**PEFT** Parameter-Efficient Fine-Tuning

**RoBERTa** Robustly Optimised BERT Approach

**SOTA** state-of-the-art

**VeRA** Vector-based Random Matrix Adaptation

**ViT** Vision Transformer

**VTAB** Visual Task Adaptation Benchmark

## A. Gradient Derivation

We derive the gradient for a 3 dimensional $(\mathbb{R}^{I \times J \times K})$ low-rank CPD update. This can be further generalised to an $n$-dimensional update. Consider a 3D low-rank update

$$\mathcal{F} = \delta\mathbf{W} = \sum_{r=1}^{R} \lambda_r^S \boldsymbol{s}_r^{(1)} \circ \boldsymbol{s}_r^{(2)} \circ \boldsymbol{s}_r^{(3)}, \tag{21}$$

where $\circ$ represents the outer product. In case of an individual rank $z$, we can omit the summation

$$\mathcal{F}_z = \lambda_z^S \boldsymbol{s}_z^{(1)} \circ \boldsymbol{s}_z^{(2)} \circ \boldsymbol{s}_z^{(3)}. \tag{22}$$

The gradient for $\lambda_z^S$ is the outer product of rank-one vectors without $\lambda_z^S$.

$$\nabla_{\lambda_z^S} \mathcal{F}_z = \boldsymbol{s}_z^{(1)} \circ \boldsymbol{s}_z^{(2)} \circ \boldsymbol{s}_z^{(3)}. \tag{23}$$

From here on, we absorb the $\lambda_z^S$ into the first factor $\boldsymbol{s}_z^{(1)}$ for ease of derivation. To derive the gradients for other factors, we use the tensor representation from (Kolda & Bader, 2009). The elements of the tensor $\mathcal{F}_r$ can be written as

$$\mathcal{F}_r = \boldsymbol{s}_{ir}^{(1)} \boldsymbol{s}_{jr}^{(2)} \boldsymbol{s}_{kr}^{(3)} \quad \forall i = 1...I, j = i...J, k = 1...K. \tag{24}$$

Gradient of $t^{th}$ element in $\boldsymbol{s}_r^{(1)}$ is given as

$$\nabla_{\boldsymbol{s}_{tr}^{(1)}} \mathcal{F}_r = \begin{cases} \boldsymbol{s}_{jr}^{(2)} \boldsymbol{s}_{kr}^{(3)}, & \text{if } i = t, \\ 0, & \text{if } i \neq t. \end{cases} \tag{25}$$

The above form is analogous to the Kronecker delta. Equation 25 is in element form of the tensor, converting it to CPD form results in

$$\nabla_{\boldsymbol{s}_r^{(1)}} \mathcal{F}_r = \boldsymbol{I}^{S1} \circ \boldsymbol{s}_r^{(2)} \circ \boldsymbol{s}_r^{(3)}, \tag{26}$$

where $\boldsymbol{I}^{S1} \in \mathbb{R}^{I \times I}$ is an identity matrix resulting from vectorisation of the Kronecker delta. Similarly, for the other two factors, the gradient can be represented as

$$\nabla_{\boldsymbol{s}_r^{(2)}} \mathcal{F}_r = \boldsymbol{I}^{S2} \circ \boldsymbol{s}_r^{(1)} \circ \boldsymbol{s}_r^{(3)}, \tag{27}$$

$$\nabla_{\boldsymbol{s}_r^{(3)}} \mathcal{F}_r = \boldsymbol{I}^{S3} \circ \boldsymbol{s}_r^{(1)} \circ \boldsymbol{s}_r^{(2)}, \tag{28}$$

where $\boldsymbol{I}^{S2} \in \mathbb{R}^{J \times J}$ and $\boldsymbol{I}^{S3} \in \mathbb{R}^{K \times K}$ are also identity matrices. Since the gradient for the factors is computed from the element-wise formulation of tensors, this is generalisable to an $n$-dimensional low-rank update. For an $n$-dimensional low-rank update, the gradient of the specific factor is replaced by its identity matrix.

## B. Effect of Initialisations

In this section, we provide an additional ablation on the effect of various initialisations of CPD factors. Table 7 presents the results of various initialisations such as random normal (Glorot & Bengio, 2010) and orthogonal (Saxe et al., 2014). As discussed earlier, we initialise the second factor as zero and evaluate on the SVHN dataset from the VTAB-1k benchmark. We observe that initialising the last two factors in MHA with orthogonal and the first factor by sampling from a normal distribution results in the best performance. We hypothesise that the initialisation of the first factor by sampling from a Gaussian distribution is crucial because $\boldsymbol{A}^{(1)} \in \mathbb{R}^{3l \times R}$ is the only factor containing information for each transformer block's low-rank update. Following this result, we initialise the factors for the feed-forward tensor similarly, with the bias term set to zero.

*Table 7.* Effect of various factor initialisations. $Z$ represents zero initialisation, $O$ denotes orthogonal initialisation (Saxe et al., 2014), and $N$ represent random normal initialisation (Glorot & Bengio, 2010). We used the SVHN dataset for our evaluation.

| $\boldsymbol{A}^{(1)}$ | $\boldsymbol{A}^{(2)}$ | $\boldsymbol{A}^{(3)}$ | $\boldsymbol{A}^{(4)}$ | Accuracy |
|---|---|---|---|---|
| $O$ | $Z$ | $O$ | $O$ | 89.79 |
| $O$ | $Z$ | $N$ | $N$ | 89.81 |
| $N$ | $Z$ | $N$ | $N$ | 89.95 |
| $N$ | $Z$ | $O$ | $O$ | **90.50** |

## C. Implementation Details

In this section, we provide details about the VTAB-1k and FGVC benchmarks, as well as the benchmark (Kopiczko et al., 2024) for fine-tuning ViT-L. For each benchmark, we provide the used hyperparameters.

### C.1. Software and Hardware Details

We use PyTorch (Paszke et al., 2017), Tensorly (Kossaifi et al., 2019) and PyTorch Image Models (Wightman, 2019) for model fine-tuning and evaluation. The pre-trained ViT-B checkpoint is available here[1]. We fine-tuned the ViT on one Nvidia GA100 GPU for the VTAB-1k benchmark and one Nvidia H100 GPU for the FGVC benchmark. For evaluation, we use an Nvidia RTX A5000. In the case of language experiments, we use a maximum of 8 Nvidia GA100 GPUs for fine-tuning and evaluation.

### C.2. VTAB-1K

The Visual Task Adaptation Benchmark (VTAB)-1k consists of 19 diverse datasets categorised into Natural, Specialised, and Structured groups. Firstly, the natural group consist of datasets from classical vision problems such as CIFAR-100 (Krizhevsky & Hinton, 2009), Caltech-101 (Fei-Fei et al., 2004), DTD (Cimpoi et al., 2014), Flowers102 (Nilsback & Zisserman, 2008), Pets (Parkhi et al., 2012), SVHN (Netzer et al., 2011), and Sun397 (Xiao et al., 2010). Secondly,

---

[1] https://storage.googleapis.com/vit_models/imagenet21k/ViT-B_16.npz

the Specialised group contains images from special sensors like satellites and microscopes. The datasets in this group are Resisc45 (Cheng et al., 2017), EuroSAT (Helber et al., 2019), Patch Camelyon (Veeling et al., 2018), and Diabetic Retinopathy (Kaggle & EyePAcs, 2015). Finally, the third group includes Clevr (Johnson et al., 2017) (location and orientation), SmallNorb (LeCun et al., 2004) (azimuth and elevation), DMLab (Beattie et al., 2016), and KITTI (Geiger et al., 2013). The Natural group of datasets contains images of domains similar to those in ImageNet. Meanwhile, the Structured and Specialised groups contain images from domains dissimilar to ImageNet (Zhai et al., 2019). This diverse mix of domains makes this benchmark suitable for evaluating the fine-tuning characteristics. We initialise $\lambda^A$ and $\lambda^B$ by sampling from a normal distribution with mean $\lambda_\mu$ and standard deviation $\lambda_\sigma$. Table 8 presents the detailed hyperparameters.

*Table 8.* Hyperparameter details for the VTAB-1k benchmark using the ViT-Base model. The standard deviation (std) is computed over 10 runs.

| Hyperparameter | Cifar100 | Caltech101 | DTD | Flower102 | Pets | SVHN | Sun397 | Camelyon | EuroSAT | Resisc45 | Retinopathy | Clevr-Count | Clevr-Dist | DMLab | KITTI-Dist | dSpr-Loc | dSpr-Ori | sNORB-Azim | sNORB-Ele |
|---|---|---|---|---|---|---|---|---|---|---|---|---|---|---|---|---|---|---|---|
| $\alpha$ | 0.1 | 100 | 0.1 | 10 | 1 | 100 | 1 | 10 | 10 | 10 | 0.1 | 5 | 2.5 | 10 | 5 | 50 | 1 | 100 | 10 |
| $\lambda_\mu$ | 1.5 | 0.9 | 1.0 | 1.0 | 1.2 | 1.0 | 1.35 | 1.0 | 1.08 | 1.16 | 1.0 | 1.0 | 1.0 | 1.0 | 1.0 | 1.0 | 1.3 | 1.0 | 1.0 |
| $\lambda_\sigma$ | 0.1 | 0.01 | 0.1 | 0.02 | 0.06 | 0.05 | 0.06 | 0.0 | 0.028 | 0.03 | 0.0 | 0.0 | 0.0 | 0.0 | 0.0 | 0.0 | 0.07 | 0.0 | 0.0 |
| rank | 32 | 32 | 16 | 32 | 16 | 32 | 16 | 32 | 32 | 32 | 16 | 32 | 16 | 32 | 32 | 32 | 16 | 32 | 16 |
| std | 0.15 | 0.31 | 0.16 | 0.06 | 0.18 | 0.18 | 0.04 | 0.31 | 0.10 | 0.19 | 0.11 | 0.12 | 0.10 | 0.26 | 0.37 | 0.73 | 0.42 | 0.37 | 0.58 |

In Table 2, the results of Linear, FT, VPT-Shallow, VPT-Deep, and Adapter-256 are from (Jia et al., 2022). The results for Adaptformer and RepAdapter are from (Luo et al., 2023), and the results for LoRA are from (Jie & Deng, 2023). The other results are from their corresponding papers.

## C.3. FGVC

The Fine-Grained Visual Categorization (FGVC) benchmark is a collection of five large datasets: CUB-200-2011 (Wah et al., 2011), NABirds (Van Horn et al., 2015), Oxford Flowers (Nilsback & Zisserman, 2008), Stanford Dogs (Khosla et al., 2011), and Stanford Cats (Gebru et al., 2017). These five datasets are purely domain-specific and focus on fine-grained categories. We use the whole training set, unlike the VTAB-1k benchmark. The validation split is done with statistics from (Jia et al., 2022) with seed 0. The class information is as follows: CUB-200-2011 contains 200 classes, NABirds 555 categories, Oxford Flowers 102 classes, Stanford Dogs 120 classes, and Stanford Cars 196 classes. The hyperparameters for the FGVC benchmark are presented in Table 9.

*Table 9.* Hyperparameter details for the FGVC benchmark using the ViT-Base model.

| Hyperparameter | CUB-200-2011 | NABirds | Oxford Flowers | Stanford Dogs | Stanford Cars |
|---|---|---|---|---|---|
| $\alpha$ | 0.01 | 0.01 | 10 | 0.001 | 100 |
| $\lambda_\mu$ | 1.0 | 1.0 | 1.0 | 0.9 | 1.0 |
| $\lambda_\sigma$ | 0.0 | 0.0 | 0.02 | 0.02 | 0.0 |
| rank | 32 | 32 | 32 | 32 | 32 |

## C.4. Fine-tuning ViT-L

Following experiments from Kopiczko et al. (2024), we fine-tune ViT-L on the CIFAR100 (Krizhevsky & Hinton, 2009), Food101 (Bossard et al., 2014), Oxford Flowers (Nilsback & Zisserman, 2008), and Resisc45 (Cheng et al., 2017) datasets. The experimental setup uses 10 training samples per class from the training set and evaluates on the test set. We use the *numpy* random choice with seed 6 for sampling to ensure reproducibility. Table 10 presents the CaRA hyperparameters for the individual datasets. Regarding this experiment, we also fine-tune other PEFT methods such as DoRA (Liu et al., 2024) and PiSSA (Meng et al., 2024). We use the implementations of these methods from (Mangrulkar et al., 2022). For DoRA

and PiSSA, we use a rank of 8 and an $\alpha$ value of 8, and fine-tune for 100 iterations.

*Table 10.* Hyperparameter details for four image classification datasets using the ViT-Large model. The standard deviation (std) is computed over five runs.

| Hyperparameter | CIFAR100 | Food101 | Flowers102 | Resisc45 |
|:---:|:---:|:---:|:---:|:---:|
| $\alpha$ | 0.01 | 0.01 | 0.01 | 0.01 |
| $\lambda_\mu$ | 1.0 | 1.0 | 1.0 | 1.0 |
| $\lambda_\sigma$ | 0.01 | 0.01 | 0.0 | 0.01 |
| rank | 32 | 32 | 32 | 32 |
| std | 0.16 | 0.33 | 0.03 | 0.19 |

## D. Additional fine-tuning results

We provide additional experiments for fine-tuning Swin Transformer (Liu et al., 2021) and the Robustly Optimised BERT Approach (RoBERTa) (Liu et al., 2023).

### D.1. Fine-tuning Swin-B

Swin Transformer (Liu et al., 2021) adds hierarchical structures to ViT and we demonstrate that CaRA works with hierarchical architectures, which use different embedding dimensions and number of heads for different layers, as well. For the experiments, we use the Swin-Base model pre-trained on ImageNet-21k. The pre-trained model checkpoint is available here[2]. The Swin-B layers are arranged into four stages, each with varying embedding dimensions $d_{\text{model}}$ and number of heads $h$. The proposed tensorisation in Section 3.2.1 must thus be adapted for each stage individually. Table 11 details each stage with corresponding tensors $\mathbf{W}^{\text{mha}}$ and $\mathbf{W}^{\text{ffn}}$ shapes.

*Table 11.* Tensorising Swin Transformer: Tensor shapes for $\mathbf{W}^{\text{mha}}$ and $\mathbf{W}^{\text{ffn}}$ across four stages of Swin-B. For each stage, the number of layers, embedding dimension ($d_{\text{model}}$), and number of attention heads ($h$) are also reported.

| Stage | #Layers | $d_{\text{model}}$ | $h$ | $\mathbf{W}^{\text{mha}}$ | $\mathbf{W}^{\text{ffn}}$ |
|:---:|:---:|:---:|:---:|:---:|:---:|
| 1 | 2 | 128 | 4 | $6 \times 128 \times 4 \times 32$ | $18 \times 128 \times 128$ |
| 2 | 2 | 256 | 8 | $6 \times 256 \times 8 \times 32$ | $20 \times 256 \times 256$ |
| 3 | 18 | 512 | 16 | $54 \times 512 \times 16 \times 32$ | $162 \times 512 \times 512$ |
| 4 | 2 | 1024 | 32 | $6 \times 1024 \times 32 \times 32$ | $22 \times 1024 \times 1024$ |

**Results.** We fine-tuned the Swin Transformer with CaRA on the VTAB-1k datasets. The results in Table 12 show that CaRA outperforms existing PEFT methods in terms of accuracy, despite using less trainable parameters than most of the other methods. While SPT-LoRA and FacT-TT perform slightly better on the Natural datasets, CaRA outperforms them on the Special and Structured datasets.

### D.2. Natural Language Understanding (NLU)

To evaluate CaRA's capabilities for the language domain, we evaluate it on the General Language Understanding Evaluation (GLUE) benchmark using the RoBERTa model. Our experimental setup closely aligns with (Hu et al., 2022). In particular, we apply CaRA only to query and value projection matrices in the MHA module. CaRA is trained with rank 64, $\alpha$ is set to 100, and all the $\lambda$'s are initialised as ones. The results for FT and non-tensor based PEFT methods are from (Hu et al., 2022). For the other methods, we use the results from their corresponding publications.

**Results.** Table 13 demonstrates the performance of CaRA on the GLUE benchmark when fine-tuned with the RoBERTa-Base model. We observe that CaRA outperforms existing tensor-based fine-tuning methods such as LoTR (Bershatsky et al., 2024) and LoRETTA$_{adapter}$ (Yang et al., 2024). We observe that tensor-based methods like LoTR and LORETTA lag in

---

[2]https://github.com/SwinTransformer/storage/releases/download/v1.0.0/swin_base_patch4_window7_224_22k.pth

*Table 12.* VTAB-1k evaluation results with the Swin-Base model. Our method results are highlighted in grey. Mean accuracy and parameters are averaged over group-wise mean values. We present the best result in bold and the second-best is underlined.

| | #Params(M) | Natural | Special | Structured | Group Mean |
|---|---|---|---|---|---|
| FT | 86.7 | 79.1 | 86.2 | 59.7 | 75.00 |
| Linear | - | 73.5 | 80.8 | 33.5 | 62.60 |
| BitFit | 0.201 | 74.2 | 80.1 | 42.4 | 65.57 |
| VPT-Shallow | **0.003** | 79.9 | 82.5 | 37.8 | 66.73 |
| VPT-Deep | 0.162 | 76.8 | 84.5 | 53.4 | 71.57 |
| LoRA | 1.023 | 81.7 | 87.2 | 60.1 | 76.33 |
| SPT-LoRA | 0.424 | **83.1** | 87.4 | 60.4 | 76.97 |
| FacT-TT | 0.135 | **83.1** | 86.9 | 62.1 | 77.37 |
| CaRA (ours) | 0.134 | 82.9 | **87.7** | **63.0** | **77.89** |

performance compared to LoRA, FT, and Adapter methods. CaRA further closes this performance gap to LoRA and full fine-tuning (FT), with only $0.04\%$ of FT parameters.

*Table 13.* GLUE benchmark evaluation results with the RoBERTa-Base model. Following (Hu et al., 2022), we report Matthew's correlation for CoLA, Pearson correlation for STS-B, and accuracy for other tasks.

| | #Params (M) | MNLI | QQP | MRPC | SST-2 | CoLA | QNLI | RTE | STS-B | Mean |
|---|---|---|---|---|---|---|---|---|---|---|
| FT | 125 | **87.6** | **91.9** | 90.2 | 94.8 | 63.6 | 92.8 | 78.7 | 91.2 | 86.4 |
| **Non-Tensor based** | | | | | | | | | | |
| Adapter | 0.30 | 87.1 | 90.2 | 88.5 | 94.2 | 60.8 | 93.1 | 71.5 | 89.7 | 84.4 |
| BitFit | 0.10 | 84.7 | 84.0 | **92.7** | 93.7 | 62.0 | 91.8 | 81.5 | 90.8 | 85.2 |
| LoRA (r=8) | 0.30 | 87.5 | 90.8 | 89.7 | **95.1** | 63.4 | **93.3** | **86.6** | **91.5** | **87.2** |
| **Tensor based** | | | | | | | | | | |
| LoTR (r=32) | 0.074 | 85.2 | 87.4 | 85.9 | 93.0 | 60.5 | 90.0 | 66.0 | 88.8 | 77.2 |
| LoTR (r=88) | 0.321 | 84.7 | 86.9 | 88.0 | 93.3 | 61.3 | 92.0 | 67.0 | 91.0 | 79.0 |
| LoRETTA$_{adapter}$ | 0.10 | 85.6 | 87.2 | 91.1 | 94.4 | 62.7 | 92.1 | 78.7 | 90.3 | 85.2 |
| CaRA (r=64)(ours) | **0.055** | 86.5 | 89.7 | 90.4 | 94.6 | **66.1** | 92.7 | 77.3 | 89.7 | 85.9 |

# E. Saliency Maps

This section provides the saliency maps using Integrated Gradients, as discussed in Section 5.3. Figure 6 shows the saliency maps for CaRA using ranks 32 and 16 for a few images of the FGVC-Aircraft dataset. This dataset contains aircraft from 70 families, such as Boeing 747, LockheedMartin TriStar L-1011, and Airbus A380. This dataset is ideal because aircraft are a class in ImageNet. A pretrained backbone is thus able to differentiate an aircraft from other real-world objects, but it cannot classify various aircraft families.

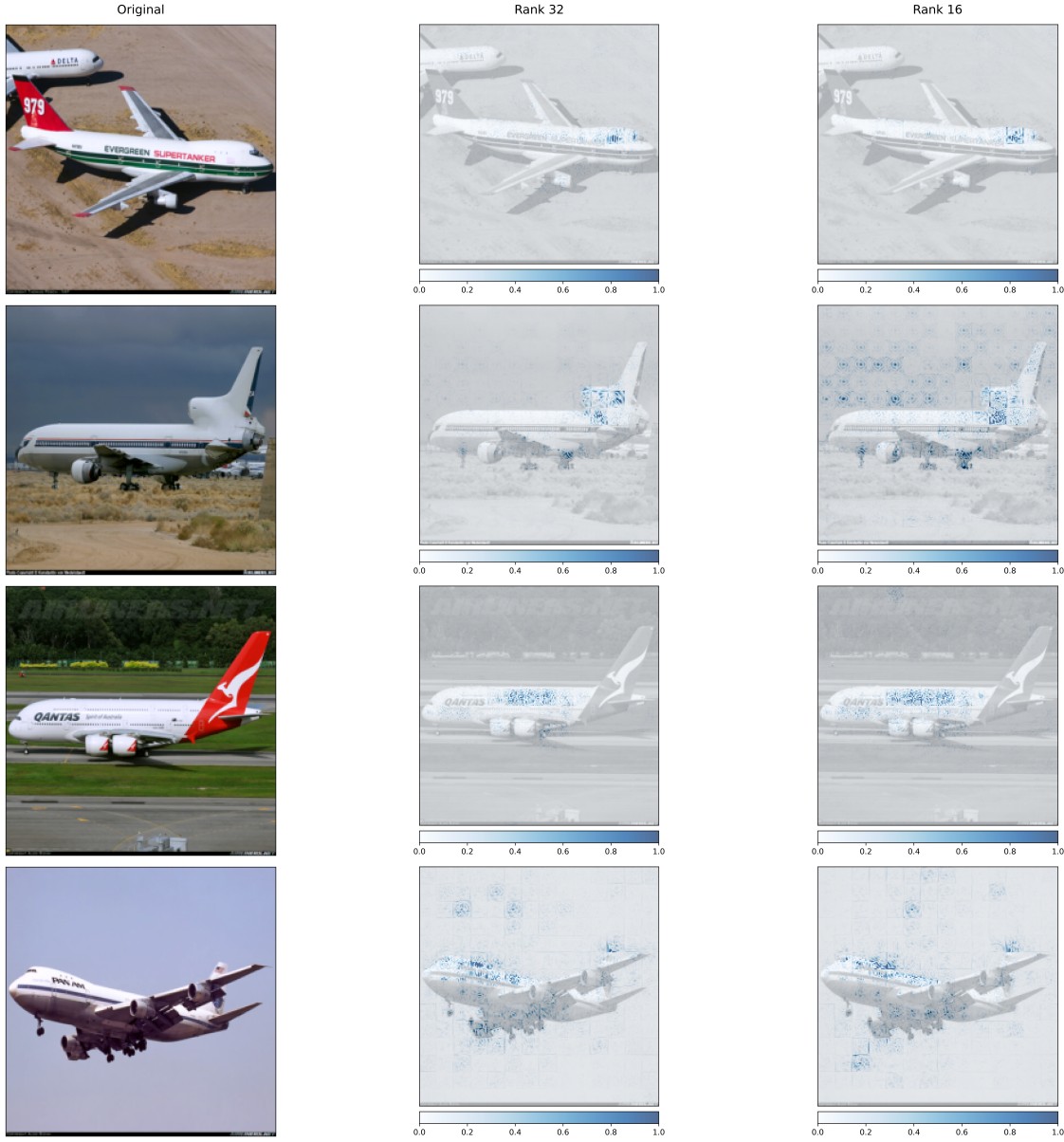

*Figure 6.* Saliency maps depicting the gradient attributions of CaRA fine-tuning ViT-B/16 for aircraft family classification, especially for ranks of 32 and 16.

