# OpenReview forum: "Canonical Rank Adaptation: An Efficient Fine-Tuning Strategy for Vision Transformers"
_ICML.cc/2025/Conference — ICML 2025 poster_

### Official Review · Reviewer_dZjR · 2025-03-06

**Overall Recommendation:** 3

**Summary:**

This paper introduces Canonical Rank Adaptation (CaRA), a peft method method specifically designed for ViTs. The core idea of CaRA is to tensorise transformer weights across layers and to directly optimize the stack using a Canonical-Polyadic Decomposition.
The authors report minimized trainable parameters and match or outperforms existing PEFT methods with lower parameter counts training a ViT-B/16 on the VTAB-1k and FGVC visual classification benchmarks.
Experimental results on these benchmarks and ablation studies are presented to support these claims.

## update after rebuttal

The authors have address most of my concern.
The increased wall time and memory of CaRA is a limitation of the method in its current form so I ask the authors to clearly include these findings in the camera ready
The approach remains novel and interesting for future research so I maintain my original score

**Claims And Evidence:**

The claims of parameter efficiency and good performance are clearly supported in the experiments.

**Essential References Not Discussed:**

A few other algorithms tackle reducing the amount of trainable parameters in LoRA. The contributions are quite orthogonal to the Canonical-Polyadic Decomposition. Examples that could be added to the related work include VeRA [3] or NoLA [4].

[3] Kopiczko, Dawid J., Tijmen Blankevoort, and Yuki M. Asano. "Vera: Vector-based random matrix adaptation." ICLR 2024
[4] Koohpayegani, Soroush Abbasi, et al. "Nola: Compressing lora using linear combination of random basis." ICLR 2024

**Experimental Designs Or Analyses:**

The proposed evaluation makes sense although I would have expected more experimental results with larger vision architectures (e.g. ViT-L/14 and above). Results for language tasks such as commonsense could also have better supported the evaluation claims.

**Methods And Evaluation Criteria:**

The proposed evaluation makes sense although I would have expected more experimental results with larger vision architectures (e.g. ViT-L/14 and above). Results for language tasks such as commonsense [1] could also have better supported the evaluation claims.

[1] Hu, Zhiqiang, et al. "Llm-adapters: An adapter family for parameter-efficient fine-tuning of large language models."  EMNLP 2023.

**Other Comments Or Suggestions:**

No other comment

**Other Strengths And Weaknesses:**

The idea of applying Canonical-Polyadic Decomposition as a parameter efficient algorithm is interesting especially as it considers the weights of the network as a whole.

Figure 1 is no very helpful in understanding the algorithm and should be improved, Figure 3 is better but there is too much $\mathbf{W}$ definitions which makes very loaded.

The experimental setting is limited to vision and one (small) network architecture. I would have liked to see experiments with CLIP or on language benchmarks to better substantiate the results and maybe get more insights as to what CaRA does differently to LoRA.

Most importantly there lacks a study of the training time of CaRA compared to alternatives (especially LoRA, SPT-LORA and FacT-TT/TK) in terms of GPU hours or wall-clock time. This would be helpful to understand the practical efficiency of CaRA beyond just parameter count.

There is no section addressing the limitations of CaRA and recommendations for future work

**Questions For Authors:**

Did the authors perform a wall-time study of training time for CaRA compared to other PEFT ? I would also be interested in whether CaRA requires more VRAM to train that the alternatives.

What are limitations of CaRA or future directions for research into PEFT with Canonical-Polyadic Decomposition ?

**Relation To Broader Scientific Literature:**

This paper is relevant to literature looking at reducing the number of parameters beyond LoRA's own reduction.

CaRA is relevant in this setting as it provide yet another alternative that appears to be competitive in terms of performance and explores new factorization ideas for PEFT. Setting ranks as global to the whole architecture and not to specific layers is also relevant as it allows for an extension to other rank-adaptive methods such as AdaLoRA [2].

[2] Zhang, Qingru, et al. "Adalora: Adaptive budget allocation for parameter-efficient fine-tuning." ICLR 2023

**Theoretical Claims:**

Gradient derivations are provided but I did not check the correctness in depth.

---

> ### Author Rebuttal · Authors · 2025-03-31
>
> Thank you for the thoughtful review and for recognising CaRA's relevance in PEFT methods. We appreciate your insights on broader evaluation and efficiency comparisons. We address the questions below in detail.
>
> ***The proposed evaluation makes sense although I would have expected more experimental results with larger vision architectures (e.g. ViT-L/14 and above).***
>
> We performed experiments on ViT-L pretrained on ImageNet-21k. Please refer to the Bbm7 reviewer's rebuttal response for the results.
>
>
> ***Results for language tasks such as commonsense [1] could also have better supported the evaluation claims.***
>
> Thank you for the suggestion. We would like to emphasise that our study focuses on image classification tasks and we do not make claims regarding language tasks. We already provide additional results for large vision transformers (ViT-L), and we expect that it works for language models as well. Due to the short time for the response, we cannot provide an evaluation for language models, but we will include them in the paper or supplementary material for the camera-ready version.
>
>
> ***Related work include VeRA [3] or NoLA [4].***
>
> Thank you for pointing out these works. We will ensure that VeRA and NoLA are included in the related work section in the camera-ready version. Additionally, we provide comparisons to the VeRA benchmark on ViT-Large in our experiments.
>
> ***Figure 1 is no very helpful in understanding the algorithm and should be improved,  Figure 3 is better but there is too much W definitions which makes very loaded.***
>
> Thank you for the comment. Figure 1 only presents the performance of CaRA. We assume the comment refers to Figure 2; we will rework Figure 2 to make it more explainable.
> We appreciate your feedback on Figure 3. We recognise that "W" definitions could be integrated directly into the figure. We will reorganise the figure to make it more intuitive and improve the layout for the camera-ready version.
>
> ***measure wall clock time and memory***
>
> As suggested, we present the wall time and VRAM allocated for various fine-tuning methods on ViT-L trained on CIFAR100 for 10 epochs.
> We observe that LoRA is the most efficient in terms of training time and memory. We attribute this speed to CUDA-optimised matrix multiplications in PyTorch.
> In contrast, CaRA shows a higher walltime because, just like the Tensorly [3] package we rely on, it is largely written in Python.  While the implementation makes it easy to use, it is not the most efficient implementation. We expect significant improvements in speed if CaRA's operations are optimized. Also, CaRA's multi-dimensional tensor nature results in slightly higher VRAM allocation.
> Given the added representational capability and the implementation, this behaviour for CaRA is to be expected. In spite of slightly higher wall time and memory, CaRA show notable performance improvements in both ViT-B and ViT-L architectures. Given that fine-tuning CaRA is often a one-time cost, we believe this is a reasonable tradeoff.
> In the case of FacT methods, we notice that they require lower ranks to match CaRA's parameter count. FacT achieves lower accuracy, with 88.4 (TK) and 87.96 (TT), while CaRA achieve 89.36.
> Interestingly, the DoRA (matrix-based) method exhibits higher memory usage and wall time, which we attribute to extra weight normalisation applied in each forward pass.
> We are still working on the experiments regarding SPT-LoRA.
>
> Method| Walltime (seconds)($\downarrow$)| VRAM (GB)($\downarrow$) |
> |-|-|-|
> LoRA|**165.7560**|**20.1079**|
> DoRA|204.0761|28.0645|
> FacT-TT|178.2826|20.2464|
> FacT-TK|180.5781|20.2443|
> CaRA(ours)|206.5548|21.3740|
>
>
> ***limitations of CaRA and future directions***
>
> This study currently focuses on vision transformers. We do not present results for other tasks, like language processing, at the moment. We expect that it works for language models as well, and we will include results for language models in the paper or supplementary material of the camera-ready version. Currently, matrix multiplications are more optimized than tensor decompositions. Using a hardware-optimized tensor decomposition will be an interesting future direction. DoRA [2] introduces a weight normalisation. A normalised form of the CP-Decomposition could further boost performance and reduce training time. We will add a discussion of limitations and future directions.
>
> References:
>
> [1] Kolda, Tamara G., and Brett W. Bader. "Tensor decompositions and applications." SIAM review 51.3 (2009): 455-500.
>
> [2] Liu, Shih-Yang, et al. "Dora: Weight-decomposed low-rank adaptation." Forty-first International Conference on Machine Learning. 2024.
>
> [3] Jean Kossaifi, Yannis Panagakis, Anima Anandkumar and Maja Pantic, TensorLy: Tensor Learning in Python, Journal of Machine Learning Research (JMLR), 2019, volume 20, number 26.

---

### Official Review · Reviewer_UspR · 2025-03-12

**Overall Recommendation:** 3

**Summary:**

This paper proposes CaRA, which uses the canonical polyadic decomposition (CPD) to replace the matrix multiplication in LoRA. There are two advantages of using CPD. Firstly, the multi-dimensional formulation can capture the structure of the head-dimension in the projection matrices in multi-head attention (MHA). Secondly, it uses fewer parameters than LoRA for the same rank. The paper also separates the matrices in MHA and those in FFN so that different numbers of decomposition dimensions can be used. CaRA is applied to ViT and tested on VTAB and FGVC benchmarks. The results are slightly better than the best baselines or on par with them.

**Claims And Evidence:**

The claim on the CaRA formulation and the parameter efficiency of CaRA is supported. Overall, the performance gain over the baselines is partially supported because the gap between CaRA and the best baselines is marginal.

**Essential References Not Discussed:**

The paper did not discuss the improved versions of LoRA, which are highly related. For example, it is helpful to compare with DoRA (ICML'24) and PiSSA (NeurIPS'24).

**Experimental Designs Or Analyses:**

The experiments on VTAB and FGVC are sound. The ablation study on changing ranks and dims is also sound. However, the visualization of heatmaps provides limited interpretability and is not convincing.

**Methods And Evaluation Criteria:**

Both the method and the evaluation make sense as the paper focuses on parameter-efficient tuning in the vision domain.

**Other Comments Or Suggestions:**

Most LoRA-related papers benchmark on LLMs, such as natural language understanding and natural language generation. It will be more comprehensive if the proposed CaRA is tested on the NLP domain.

**Other Strengths And Weaknesses:**

Clarity:

It is unclear how the baseline results in Tab.2 are obtained. Are they trained by the authors or referenced from other papers? Specifically, how much efforts are taken to tune hyper-parameters of baselines. For example, are the learning rate and the alpha value fairly tuned for LoRA? Besides, it would be helpful to readers if the rank of LoRA/CaRA were listed in the table.

Significance:

From the results of Tab.4, it seems that making the heads in MHA an extra dimension does not offer much performance improvement. So, the benefit of exploiting the head dimension is overrated in the introduction.

**Questions For Authors:**

1. L196-199. "Contrary to the existing works, this formulation ... capture any relations across heads ...". Why cannot LoRA capture relations across heads?
2. There are several notations to clarify. Firstly, what does the $[...]$ mean in Equ. (3-7)? concatenation/stacking? Secondly, what does the {$... ; ...$} mean in Equ. (8)?

**Relation To Broader Scientific Literature:**

The paper relates to the parameter-efficient fine-tuning of large models. Previous research shows that LoRA is effective, and the idea in this paper provides a similar but alternative formulation using the CP decomposition of tensors.

**Theoretical Claims:**

There are no theoretical claims.

---

> ### Author Rebuttal · Authors · 2025-03-31
>
> Thank you for your thoughtful review and for finding our experimentation sound. Below are our responses to the points raised in the review.
>
> ***performance baseline marginal***
>
> Considering SPT-LoRA as the best baseline in both VTAB-1k and FGVC benchmarks, we want to highlight that with only $\approx 11%$ of SPT-LoRA's parameters, CaRA achieves this performance.
> Furthermore, compared to other tensor-based methods, FacT-TK and FacT-TT, as shown in Figure 4, CaRA consistently achieves $\approx 1%$ or more accuracy in all three types of VTAB datasets, which is a substantial improvement on these benchmarks. Additionally, the ViT-L experiments presented below further establish CaRA's strong performance and scalability. The gains compared to the other approaches are even larger for ViT-L. Overall, these results demonstrate CaRA's effectiveness in fine-tuning the vision transformer across multiple benchmarks.
>
> Method|ViT-L #Params ($\downarrow$)|CIFAR100 ($\uparrow$)|Food101 ($\uparrow$)|Flowers102 ($\uparrow$)|Resisc45($\uparrow$)|Mean ($\uparrow$)
> |-|-|-|-|-|-|-|
> Head | -  | 79.4  | 76.5 |98.9|67.8|80.65
> Full |  303.3M | 86.8 |78.7|98.8|79.0|85.83
> LoRA |  786.4K | 87.0|79.5|99.1|78.3|85.98
> VeRA | **61.4K** | 87.5 |79.2|99.2|78.6|86.13
> PiSSA |  786.4k | 87.11|79.55|**99.72**|78.55|86.24
> DoRA |  860.2K | 87.93|81.15|99.57|80.33|87.25
> CaRA (ours) | 75.6K | **89.36** | **83.65** |99.63|**82.43**|**88.77**
>
> ***heatmap interpretability***
>
> We use the integrated gradient maps as a tool to interpret the model's behaviour during fine-tuning, particularly by highlighting the influential image regions the model relies on.
> While we acknowledge that these visualisations alone may not provide a complete explanation, they serve as an initial step towards understanding the model's decision-making process.
>
> ***comparison to DoRA and PiSSA***
>
> Thanks for pointing out. We have included the comparisons with DoRA and PiSSA for ViT-Large training, as detailed in response to Reviewer Bbm7. We will also incorporate these comparisons in the camera-ready version and extend additionally  the discussion in the related work section.
>
> ***unclear baseline results in Tab.2***
>
> The baseline results for LoRA are from FacT [1], where the LoRA rank is set to 8. The results of Adapter and Adaptformer are from RepAdapter [3], and the other results are from their respective papers. For CaRA, we provide the details of the hyperparameters in the supplementary.
> To enhance clarity, we will update Table 2 to explicitly state the ranks for each method and include the sources of the baseline results.
>
>
> ***Tab 4, MHA extra dimension***
>
> Table 4 presents an ablation study on CP-Decomposition across multiple dimensions. When comparing rows 2 and 3 in Table 4, we observe that both the number of parameters decreases and the accuracy increases slightly with d_h as extra dimension. The main gain is in the reduction of parameters. We will clarify this in the camera-ready version.
>
>
> ***NLP domain.***
>
> Thank you for the suggestion. We would like to emphasise that our study focuses on image classification tasks and we do not make claims regarding language tasks. We already provide additional results for large vision transformers (ViT-L), and we expect that it works for language models as well. Due to the short response time, we cannot provide an evaluation for language models, but we will include them in the paper or supplementary material for the camera-ready version.
>
> ***Why cannot LoRA capture relations across heads?***
>
> Lora works with stacks of two-dimensional matrices for the fine-tuning process. By definition, two-dimensional matrix structures and their decompositions model two-dimensional relationships.
> Transformer networks are defined to have the embedding dimension (d_model), the number of heads (n_h) and the dimension of each head (d_h) [1]. If we stack multiple layers, we end up with a four dimensional tensor. Matrix-based structures do not allow us to model tensors directly. Tensor decompositions solve this issue by providing a potentially n-dimensional structure [2]. This paper leverages the Canonical Decomposition, which was designed to handle the tensor data structure that Transformers create naturally.
>
> ***Notations to clarify***
>
> Thank you for pointing it out, "[...]" in equations (3-6) corresponds to stacking, while equation 7 represents concatenations.
> We followed the exact representation as in [2]. {} represents the set of CP-Decomposition factors in matrix form. We will update the camera-ready version accordingly.
>
> References:
>
> 1.  Vaswani, Ashish, et al. "Attention is all you need." Advances in neural information processing systems 30 (2017).
>
> 2.  Kolda, Tamara G., and Brett W. Bader. "Tensor decompositions and applications." SIAM review 51.3 (2009): 455-500.
>
> 3.  Luo, Gen, et al. "Towards efficient visual adaption via structural re-parameterization." arXiv preprint arXiv:2302.08106 (2023).

---

### Official Review · Reviewer_Bbm7 · 2025-03-13

**Overall Recommendation:** 3

**Summary:**

This paper introduces Canonical Rank Adaptation (CaRA), an efficient fine-tuning strategy for Vision Transformers (ViT). The key finding is that leveraging tensor mathematics can effectively address the high-dimensionality of Multi-Head Attention (MHA), enhancing fine-tuning performance. The main results demonstrate that CaRA outperforms existing methods in visual classification benchmarks such as VTAB-1k and FGVC, while using fewer trainable parameters. The core algorithmic idea is to quantize the Transformer into two tensors, which are used for the MHA projection layer and the feedforward layer respectively, and then fine-tune with low-rank updates in the form of Canonical Polyadic Decomposition (CPD).

## update after rebuttal:

The rebuttal of the author explained most of the problems. Although this method is novel, its computational weight is not superior, so I think the original score of 3 should be maintained

**Claims And Evidence:**

The proposed claim is supported by derivational proof and citations.

**Essential References Not Discussed:**

This method bears some similarity to the FacT method, and FacT should be introduced in the related work section rather than solely when comparing methods.

**Experimental Designs Or Analyses:**

1.The method has not been fine-tuned and tested on larger models such as ViT-L or ViT-H, so it cannot be proven whether it can maintain high accuracy on these larger models.
2.When calculating the experimental mean, it is not reasonable to first compute the average accuracy for the three datasets of Natural, Specialized, and Structured, and then calculate the average of these averages. The correct approach should be to average the results across all 19 datasets. Therefore, the accuracy of the CaRA method should be 74.14%.

**Methods And Evaluation Criteria:**

The proposed method and evaluation criteria are meaningful for visual classification problems. Tensorizing the Transformer and using CPD (Canonical Polyadic Decomposition) for low-rank updates fully consider the high-dimensional characteristics of MHA (Multi-Head Attention), enabling more efficient feature capture.

**Other Comments Or Suggestions:**

None.

**Other Strengths And Weaknesses:**

Strength:
The language is fluent and the article is easy to read.
This method is innovative, and the paper provides mathematical derivation for the method.
Weakness:
There are deficiencies in the design of the experiment, which cannot effectively demonstrate the performance of the method.

**Questions For Authors:**

1. What is the effect of this method on language models?
2. During the fine-tuning process, multiplication turns into a 3-dimensional matrix multiplication. Will the computational load become excessively large?

**Relation To Broader Scientific Literature:**

In terms of tensor representation, existing research on tensor decomposition for fine-tuning was referenced, and based on this, a new tensorization and low-rank update method was proposed, which improves upon the deficiencies of existing methods in dealing with the high-dimensional nature of MHA.

**Theoretical Claims:**

The paper provides a detailed explanation of the gradient derivation for CaRA.

---

> ### Author Rebuttal · Authors · 2025-03-31
>
> Thank you for the insightful review and for recognising the innovation in our method and finding it meaningful for the vision classification problem. We appreciate your positive feedback. Below are the responses to your review.
>
> ***The method has not been fine-tuned and tested on larger models such as ViT-L or ViT-H, so it cannot be proven whether it can maintain high accuracy on these larger models***
>
> Following the ViT-L benchmark from VeRA [1], we evaluate our proposed CaRA method against other low-rank fine-tuning methods on four datasets. We use one A100 GPU for fine-tuning. For this experiment, we performed hyperparameter sweeps on lr, scale ($\alpha$), and various schedulers. For reproducibility, the camera-ready version will include more details of hyperparameters for CaRA, PiSSA, and DoRA, and the code will be available upon acceptance. The rank for LoRA, DoRA, and PiSSA is 8, and the rank for VeRA and CaRA is 256 and 64, respectively.
> Note: The benchmark in [1] only evaluates LoRA and VeRA. We also trained and evaluated PiSSA and DoRA at the request of the other reviewers.
>
> Method     |ViT-L #Params ($\downarrow$)|CIFAR100 ($\uparrow$)|Food101 ($\uparrow$)|Flowers102 ($\uparrow$)|Resisc45($\uparrow$)|Mean ($\uparrow$)
> |-|-|-|-|-|-|-|
> Head | -  | 79.4|76.5|    98.9     |   67.8     |   80.65
> Full |  303.3M |86.8    |   78.7    |    98.8     |   79.0     |   85.83
> LoRA |  786.4K |87.0    |   79.5    |    99.1     |   78.3     |   85.98
> VeRA | **61.4K** |87.5    |   79.2    |    99.2     |   78.6     |   86.13
> PiSSA |  786.4k |   87.11   |   79.55   |  **99.72**  |   78.55    |   86.24
> DoRA |  860.2K |   87.93   |   81.15   |    99.57    |   80.33    |   87.25
> CaRA (ours) | 75.6K     | **89.36** | **83.65** |    99.63    | **82.43**  | **88.77**
>
> The table demonstrates that, overall, CaRA significantly outperforms existing baseline methods with only a smaller fraction of trainable parameters ($\approx 10%$ of LoRA's parameters). In the Flowers102 dataset, PiSSA performs better than CaRA by only a slight margin. Additionally, CaRA achieves state-of-the-art accuracy on the CIFAR100, Food101 and Resisc45 datasets. The accuracy gains combined with CaRA's efficiency demonstrate that CaRA also maintains high accuracy for fine-tuning larger vision transformers.
>
> ***When calculating the experimental mean, it is not reasonable to first compute the average accuracy for the three datasets of Natural, Specialized, and Structured, and then calculate the average of these averages.***
>
> We follow the approach used in prior works [2,3] to maintain consistency with the literature. However, we understand your point and are happy to add one more column to the table with the overall mean.
>
> ***This method bears some similarity to the FacT method, and FacT should be introduced in the related work section rather than solely when comparing methods.***
>
> Thank you for the suggestion. We introduce and cite FacT in line 123 of the related work section, but we do not explicitly mention the name. We will update the section and introduce FacT by its name in the camera-ready version.
>
> ***What is the effect of this method on language models?***
>
> Thank you for the suggestion. We would like to emphasise that our study focuses on image classification tasks and we do not make claims regarding language tasks. We already provide additional results for large vision transformers (ViT-L), and we expect that it works for language models as well. Due to the short response time, we cannot provide an evaluation for language models, but we will include them in the paper or supplementary material for the camera-ready version.
>
> ***During the fine-tuning process, multiplication turns into a 3-dimensional matrix multiplication. Will the computational load become excessively large?***
>
> As shown in the table below, CaRA's training time is similar to DoRA and higher than LoRA. However, this increase in computational effort can be mainly attributed to the Python implementation of the Tensorly package. While the implementation makes it easy to use, it is not the most efficient implementation. A more efficient implementation could follow [4]. Nevertheless, the computational load remains small. In terms of memory usage, CaRA performs similarly to FacT.
>
> Method | Walltime (seconds)($\downarrow$) | VRAM (GB)($\downarrow$) |
> |-|-|-|
> LoRA | **165.7560** | **20.1079** |
> DoRA | 204.0761 | 28.0645 |
> FacT-TT | 178.2826  | 20.2464 |
> FacT-TK | 180.5781 | 20.2443 |
> CaRA(ours) | 206.5548 | 21.3740 |
>
> References:
>
> [1] Kopiczko, Dawid J., Tijmen Blankevoort, and Yuki M. Asano. "Vera: Vector-based random matrix adaptation." ICLR 2024.
>
> [2] Dosovitskiy, Alexey, et al. "An image is worth 16x16 words: Transformers for image recognition at scale." ICLR 2021.
>
> [3] Jia, Menglin, et al. "Visual prompt tuning." ECCV 2022.
>
> [4] Yang, Zi, Junnan Shan, and Zheng Zhang. "Hardware-efficient mixed-precision CP tensor decomposition." arXiv preprint arXiv:2209.04003 (2022).

---

### Decision · Program_Chairs · 2025-05-01

**Decision:**

Accept (poster)

**Comment:**

This paper received three reviews, with overall recommendations leaning toward weak acceptance. Reviewers raised several initial concerns, including: deficiencies in experimental design, unclear baseline results in Table 2, ambiguity around the extra dimension in MHA as shown in Table 4, lack of clarity in Figures 1 and 3, absence of experiments with CLIP or VLLM, insufficient analysis of CaRA’s training time, and limited discussion of CaRA’s potential limitations. The rebuttal addressed many of these issues, at least partially, and some improvements were committed to be included in the final version.

The AC agrees with the reviewers’ assessments and appreciates the paper’s innovative approach and solid mathematical formulation. Additionally, the experimental results, though still with room for improvement, provide important evidence of the method’s effectiveness.

The AC leans toward recommending acceptance, contingent on the authors fully addressing the outstanding concerns and incorporating all committed revisions and clarifications into the camera-ready version.